# Technical note: High density mapping of regional groundwater tables with steady-state surface nuclear magnetic resonance – three Danish case studies

Mathias Vang[1], Denys Grombacher[1,3], Matthew P. Griffiths[2], Lichao Liu[1], and Jakob Juul Larsen[2,3]

[1]Department of Geoscience, Aarhus University, Denmark
[2]Department of Electrical and Computer Engineering, Aarhus University, Denmark
[3]Aarhus Centre for Water Technology, Aarhus University, Denmark

**Correspondence:** Mathias Vang (mva@geo.au.dk)

**Abstract.**

Groundwater is an essential part of the water supply worldwide and the demands on this water source can be expected to increase in the future. To satisfy the needs and ensure sustainable use of resources, increasingly detailed knowledge of groundwater systems is necessary. However, it is difficult to directly map groundwater with well-established geophysical

methods, as these are sensitive to both lithology and pore fluid. Surface nuclear magnetic resonance (SNMR) is the only method with direct sensitivity to water and it is capable of non-invasively quantifying water content and porosity in the subsurface. Despite these attractive features SNMR has not been widely adopted in hydrological research, the main reason being an often-poor signal-to-noise ratio, which leads to long acquisition time and high uncertainty on results. Recent advances in SNMR acquisition protocols based on a novel steady-state approach has demonstrated the capability of acquiring high quality

data much faster than previously possible. In turn, this have enabled high-density groundwater mapping with SNMR. We demonstrate the applicability of the new steady-state scheme in three field campaigns in Denmark where more than 100 SNMR soundings with approximately 30 m depth of investigation were conducted. We show how the SNMR soundings enables us to track water level variations at the regional scale and we demonstrate a high correlation between water levels obtained from SNMR data and water levels measured in boreholes. We also interpret the SNMR results jointly with independent transient

electromagnetics (TEM) data, which allows us to identify regions with water bound in small pores. Field practice and SNMR acquisition protocols were optimized during the campaigns, and we now routinely measure high-quality data on eight to ten sites per day with a two-person field crew. Together, the results from the three surveys demonstrate that with steady-state SNMR it is now possible to map regional variations in water levels with high quality data and short acquisition times.

## 1 Introduction

As water scarcity increases further understanding of subsurface hydrology is crucial (Postel, 2000; Döll et al., 2009; Liu et al., 2017). Quantifying subsurface structures have historically been through borehole drilling yet the cost of drilling is high and only gives point information. To further characterize aquifer structure, geophysical methods are used which can measure

parameters of subsurface structures on a scale not feasible with boreholes (Oldenburg and Li, 2005). Additionally, geophysical methods are non-invasive.

Currently, electromagnetic (Danielsen et al., 2003; Siemon et al., 2009) and galvanic methods (Goldman and Neubauer, 1994; Mastrocicco et al., 2010) are widely in use for groundwater exploration due to their ability to densely map the subsurface. Data from these methods are linked to both pore fluid and matrix properties of the subsurface (Robinson et al., 2008). However, this dual sensitivity makes it difficult to directly quantify water in pores (Behroozmand et al., 2012).

Alternatively, surface nuclear magnetic resonance (SNMR) can directly measure the water content in the subsurface (Yara-
manci et al., 2000). SNMR acquisitions are also sensitive to pore sizes, which affects the relaxation time of the NMR signal (Yaramanci et al., 1999; Legchenko et al., 2002). Direct knowledge of pore parameters is crucial to expand hydrological insights. The SNMR method has been used to characterize aquifers in many different environments (Yaramanci et al., 2002; Behroozmand et al., 2017). SNMR, however, suffers from low amplitude signals which are often overwhelmed by electromagnetic noise. To overcome this, multiple measurements are typically stacked, leading to long acquisition times. Advances in
noise mitigation, such as remote reference noise cancellation and model based subtraction, has enabled handling of data from noisier environments (Walsh, 2008; Larsen and Behroozmand, 2016), but noise still remains an issue at many sites. Recently, rapid acquisition of high-quality data using steady-state methods was demonstrated by Grombacher et al. (2021). Steady-state SNMR can sample the subsurface much faster than standard SNMR measurements, which enables mapping of larger areas and sites where noise has previously been prohibiting.

The scope of this paper is to demonstrate high density mapping of groundwater systems using steady-state SNMR. We do this using data from three Danish field campaigns conducted in glacial landscapes. In the three campaigns we collected 29, 38, and 50 SNMR soundings. We extract estimates of water levels and water contents from the NMR data and compare the results against transient electromagnetic (TEM) measurements and borehole data. In all three cases we observe good agreement between the NMR derived results and the independent measurements. The high data density allows us to track water level
changes at larger spatial scales.

The paper is structured as follows. A brief theory section introduces the methods, followed by data collection and inversion of the data. Further, a result and interpretation section are presented for the three field campaigns. A general discussion summarizes key takeaways from this study.

## 2  Methods

### 2.1  SNMR

Surface nuclear magnetic resonance (SNMR) utilizes the magnetic moment of hydrogen nuclei in water molecules (Hertrich et al., 2008). When nuclei with a magnetic moment are placed in a static magnetic field, their magnetic moment preferentially align with this field creating a small magnetization. By transmitting an excitation pulse, with a resonant frequency, the nuclei are perturbed, shifting the magnetization from equilibrium (Yaramanci et al., 2000). The resonant frequency, called the Larmor
frequency, is proportional to the strength of the background magnetic field, which in SNMR is the Earth's magnetic field.

The excitation pulse is transmitted using a coil at the surface. When the pulse terminates, the magnetization decays towards equilibrium producing a signal, which is inductively measured by a receiver coil at the surface. The standard measurement is the free induction decay (FID), where the signal following a single excitation pulse is measured. Averaging of data from multiple pulses is necessary to improve the signal to noise ratio, but consecutive pulses are often spaced 2 s to 5 s apart to ensure the system has returned to equilibrium before the next measurement. The amplitude of the received signal is proportional to subsurface water content. Relaxation times are linked to pore size, and give insight into hydraulic conductivity (Knight et al., 2012). By increasing the current transmitted, deeper layers of the subsurface can be probed. A distinction is made in SNMR between bound and mobile water. Bound water, as in clays, leads to very short relaxation times, which are not generally measurable by SNMR, whereas the relaxation times of mobile water in sand or gravels are longer and more readily observed.

In this study we use a recently developed acquisition style for SNMR called steady-state for rapid acquisitions of high-quality data (Grombacher et al., 2021). The method consists of transmitting a long pulse-train with closely spaced pulses, typically separated by $\sim 100$ ms, which drive the magnetization into a steady-state. This differs from the FID, where the system fully returns to equilibrium before the next transmitter pulse. The steady-state equilibrium depends on the pulse-train parameters, such as pulse repetition time and pulse duration, as well as the properties of the subsurface. By rapidly measuring the NMR signal between pulses a large amount of data are acquired in less time compared to the FID approach. The results of these acquisitions can be inverted for water contents, and relaxation parameters $T_2^*$ and $T_2$, whereas only $T_2^*$ is measured with FID. Increases in stacking rates improve the signal-to-noise ratio, enabling more soundings per day, and higher density mapping. Furthermore, the steady-state approach can also shift the NMR signal away from narrow band noise sources (Grombacher et al., 2022). This is particularly useful in cases where the Larmor frequency is close to or coincides with a power line harmonic frequency.

## 2.2 SNMR instrument

We use a SNMR system, called Apsu (Larsen et al., 2020), for the acquisitions in this work, see Fig. 1. It consists of a main controller unit (TxC) which measures the NMR signal, a transmitter unit (Tx), a capacitor bank (ps), with 600 V maximum potential, and a current probe measuring the transmitted current. A 1 kW generator is used for continuously charging the capacitor bank. The generator is connected to the power supply using a 25 m extension cord to ensure that the electromagnetic noise from the generator is minimized in the receiver coil. We use a square 50 m $\times$ 50 m coincident loop to transmit and receive the NMR signal which provides a depth of investigation around 25 m to 30 m. The system can be easily mounted on two backpacks as seen in Fig. 1 and carried between sites by a two-person crew. Typical data collection rates approach 8 to 10 soundings per day with a standard acquisition scheme.

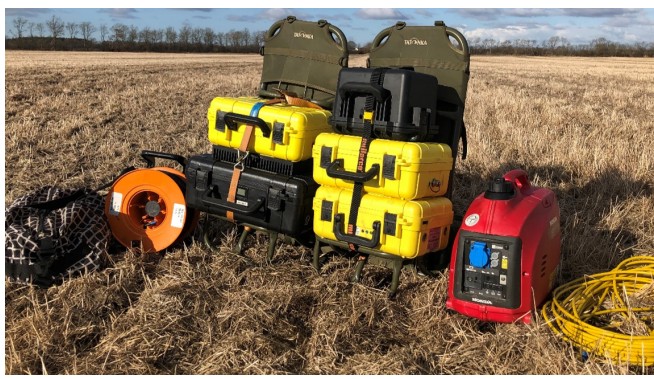

**Figure 1.** The Apsu SNMR instrument, mounted on two backpacks carried by a two person crew.

## 2.3 Data collection

The steady-state pulse trains are defined by the following parameters, pulse duration $\tau$, repetition time, style, offset, Q's, and stack number (Griffiths et al., 2022). The duration of each pulse in the pulse train, $\tau$, is between 10 ms to 100 ms. The repetition time, i.e., the amount of time between each identical pulse, is defined by an integer multiple of periods of the transmitter frequency to ensure phase coherency between pulses. These repetition times are denoted by the number of periods when oscillating at the Larmor frequency (i.e., a 200 period pulse is 200 oscillations at the transmit frequency or in Denmark around 93 ms). The style of the pulse can be either "regular", where the polarity of the pulses is the same for the full pulse train, or "alternating" where the polarity changes for each pulse (Grombacher et al., 2022). The offset, $\delta f_T$, can be used to project the signal away from noisy frequency bins (Grombacher et al., 2022). The number of pulse moments (Q's) defines how dense the current range is sampled. In this study, a current range from 5 A to 80 A was acquired for high resolution of the shallow subsurface. By increasing the current amplitude, the spatial sensitivity is changed from shallow to deep as in regular SNMR. The number of stacks is how many times each measurement is repeated at the same pulse moment. In this study, the number of stacks is chosen based on 1 minute of acquisition considering the associated repetition time. The 1-minute acquisition asserted a high data quality in Aars while maintaining a short measurement time at each site and was preserved for the following campaigns for easy comparison.

Multiple pulses with varying pulse parameters are used to fully resolve the subsurface. The pulses are sampled with the same current range, 5 A to 80 A. Pulse protocols and current amplitudes are varied to encode both spatial and relaxation time information in the collected data set. Variable current amplitudes are used to manipulate the depths of origin of the signal, while relaxation time information is encoded through manipulation of the repetition time. This is because varying the repetition time alters the induced steady-state amplitude, which is based on the underlying relaxation times (Griffiths et al., 2022). The depth of investigation is about 30 m for the soundings constrained by the coil size and maximum current.

In Table 1 general information regarding the campaigns is shown. Pulse sequences are updated between campaigns in particular, a decrease in the current sampling increases the acquisition rate while maintaining high-quality data. Tables 2, 3, and 4 describes the pulse sequences for each of the campaigns.

Standard processing schemes are used for the SNMR measurements (Kremer et al., 2022). Processing includes despiking and power line harmonic removal. Furthermore, a spectral analysis approach based on the discrete Fourier transform is used to retrieve the NMR signal from the time series (Liu et al., 2019).

The three field campaigns are conducted at sites previously mapped by a towed-TEM (tTEM) system (Auken et al., 2019). Results from the tTEM campaigns are used to identify structures in the subsurface for comparison with the SNMR water content profiles in the result section. The resistivity structure from the closest TEM sounding is used for the NMR data inversion as described in the following section. Borehole data are extracted from the Danish national database Jupiter (Hansen and Pjetursson, 2011). Water table measurements are one year to several decades old. It is plausible that these water tables have varied considerably by extraction. However, the consistency of these water table measurements across the borehole database suggests a relatively stable system throughout the years. By reproducing water table estimates consistent with available borehole data, we demonstrate the ability of surface NMR to reliably estimate the water table surface.

**Table 1.** Field campaigns overview.

| Campaign | No. sites | No. field days | No. measurements per site | Average misfit |
|---|---|---|---|---|
| Aars | 29 | 10 | 64 | 0.75 |
| Sunds | 38 | 12 | 48 | 0.80 |
| Kompedal | 50 | 5 | 25 | 1.02 |

## 2.4 Inversion

The SNMR data at each site are inverted for water content, $T_2^*$, and $T_2$. we utilize a fast-mapping approach, where data kernels are calculated in advance (Griffiths et al., 2022). The kernels are discretized by a 26-layer model with increasing thickness at depth from 0.5 m to 5.0 m, to a total depth of 50 m. We used the same starting model for all three field campaigns. The starting model is set as a half-space with 10 % water content, 0.1 s $T_2^*$ and $T_2$ set to 0.11 s. Vertical constraints of 10 %, which penalizes based on layer differences exceeding a 10 % variation, was used in all three campaigns. A stabiliser function is used in the inversion to ensure convergence (Grombacher et al., 2017). In this study a L2 stabiliser function (L2-norm) is chosen, which gives smooth inversion results. Nearby TEM derived resistivity profiles are used to construct SNMR kernels. All inversions are made using Aarhusinv (Auken et al., 2015).

## 2.5 Estimating water table

In this study we use SNMR to map water tables on a regional scale and our inversion are constrained to many-layered models. To identify the water table from the smooth regularized model, we use the peak of the water content derivative, as the largest gradient likely correspond to the transition from low to high saturation, i.e., the water table. Other regularization schemes such as blocky (L1) or sharp (minimum gradient support) (Grombacher et al., 2017) were implemented, but found to give very simi-

lar water table estimates, as the L2-norm. Layered inversions are not currently possible with the fast-mapping framework used
for the steady-state scheme, yet future research will focus on implementing this to help identify sharp structural boundaries.

## 3    Results

### 3.1    Aars field site

The first campaign was conducted in Aars, Northern Jutland and consists of ten field days with 29 soundings each separated by
100 m to 200 m. Sites are located in a rural area of 1 km$^2$ with several farms and limited infrastructure. Soundings are acquired
in agricultural fields with large power lines visible south of the area, which is the region's main noise source. The setting is a
glacial dominated geology consisting of tills, with a fluctuating sand content. Sparse borehole coverage indicates fluvial melt
water sands in parts of the area. A Paleogene clay is underlying the glacial deposits located at depths of 60 m to 50 m (Borehole
ID: 40.1006, 48.1171 (GEUS, 2023)). TEM resistivities reveal that the clay is sloping towards the surface when moving east
in the area approaching 10 m depth. Topography in the area is generally defined by a gentle dip towards south-east, where a
small stream is present. The water table from boreholes is ranging from 1 m to 10 m depth dipping towards north-west. The
pulse sequences used for the Aars field campaign is shown in Table 2. We acquire 16 pulse moments for each pulse sequence
yielding approximately one hour of acquisition per site.

**Table 2.** Pulse sequences for Aars field site.

| Pulse protocol | $\tau$ [ms] | No. periods | Style | $\delta f_T$ [Hz] | Q's | No. stacks |
|---|---|---|---|---|---|---|
| 1 | 10 | 200 | Alt | 0 | 16 | 650 |
| 2 | 10 | 200 | Reg | 0 | 16 | 650 |
| 3 | 10 | 400 | Reg | 0 | 16 | 325 |
| 4 | 40 | 800 | Alt | 0 | 16 | 162 |

Figure 2a shows the water level estimates from SNMR soundings and borehole measurements. The SNMR sites serve as
infill for the boreholes and covers the area in densely gridded estimates of water levels. The water table elevation ranges from
11 m to 22 m in the area. Topography changes account for most of this change as elevation increases towards north-west.
Additionally, a depression in water levels is found centrally in the area coinciding with a minimum in topography, which is not
sampled by the few boreholes. Some differences between SNMR and boreholes exist, especially in the north part of the area.
Here, the SNMR data identify a deeper water table (about 2 m) than the nearby borehole measurement. This could arise from
seasonal changes in water table since the boreholes are done 1.5 year prior to the SNMR soundings. The SNMR results identify
the physical location of water at depth and not the hydraulic head, as such if the aquifer is confined, there will be differences
between SNMR estimated water tables and the pressure head from wells. Borehole data in this area reveals a clay till which

could serve as the confining layer (Borehole ID: 40.2055 (GEUS, 2023)). At S6 and S7, high water contents were found close to the surface and the water table is set to the topography in these soundings.

In Fig. 2b tTEM derived resistivities and SNMR water content profiles are shown in a west to east profile. A nearby borehole is projected on to this profile with its water level measurement. Here, an upper unit of 100 $\Omega$m to 200 $\Omega$m defines the area, indicative of the sandy till seen in boreholes. The fluctuating resistivity can be linked to a varying clay content in the tills. A conductive unit underlies this till, which is interpreted to be a Paleogene clay. The SNMR water level varies mostly with topography about 12 m over the profile. Water contents in the saturated aquifer ranges from 15 % to 20 %, which correlate to the sandy till. For the SNMR soundings labeled S6 and S7 in Fig. 2b, high water contents are found at 0 m to 3 m, with a drop in water contents with increasing depth. This is interpreted to be a thin sand layer with higher mobile water contents and an underlying clay rich till with low mobile water contents. The TEM profile identifies a layer of 90 $\Omega$m to 60 $\Omega$m which is consistent with Danish tills. The shallow resistive sand layer is difficult to resolve with TEM. Most of the profiles show a decrease in water content going deeper, which correlates with a more conductive part of the subsurface, interpreted as a more clay rich part of the till overlying the conductive Paleogene clay.

## 3.2 Sunds field site

The second campaign took place near Sunds, Central Jutland and consists of 38 soundings in twelve days, Table 1. The area of interest is 12 km$^2$ with infrastructure and houses in the vicinity. Data acquisition nearby houses were a challenge, due to the difficulty of handling the noise distorting the data. Additionally, buried power lines are present in the area. This has led to data from several sites deemed unusable due to very low signal to noise ratio. Sunds is located close to the Weichselian ice margin, which has deposited a thick coarse melt water sand package. Sunds has previously been mapped by a combined tTEM and FloaTEM campaign (Maurya et al., 2022). High water contents are expected in this area defined by coarse deposits. The flat meltwater plain yields a very flat terrain varying a couple of meters over the entire area. Densely spaced boreholes cover the area, many of them used for irrigating agricultural fields. The geomagnetic field strength equals a Larmor frequency of approximately 2150 Hz where a powerful harmonic of the power lines resides. To mitigate this distortion a frequency offset is used in all regular pulses, shown in Table 3. Using offsets with regular sequences along with on resonance alternating pulses, enables the production of high-quality measurements despite the overlapping power line harmonic (Grombacher et al., 2022). The number of measurements is decreased and only 13 pulse moments are collected for each pulse.

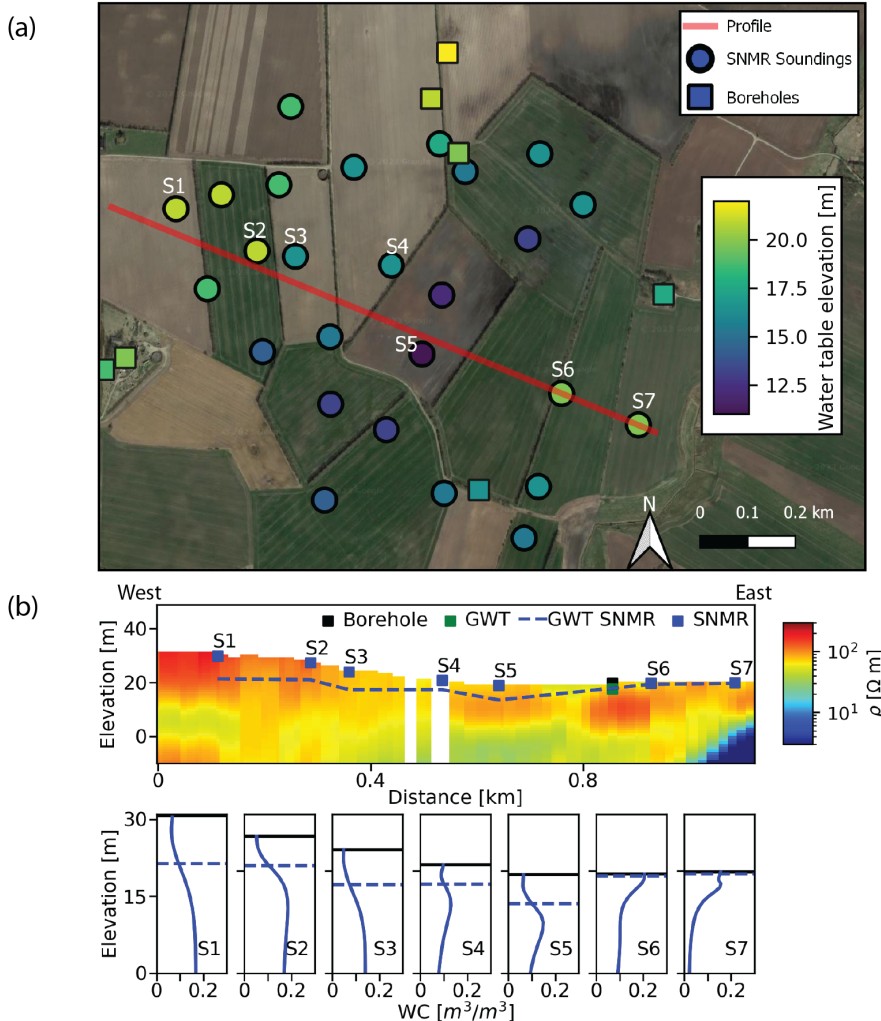

**Figure 2.** (a): Overview image of soundings(circles) and boreholes(squares) from Aars field site. The color denotes the water table elevation on the same scale for boreholes and SNMR. Profile shows the location of cross-section in (b). Map data: © Google Maps 2021. (b): Cross-section with tTEM resistivities, SNMR water content profiles and with groundwater table (GWT) estimates from SNMR and boreholes. Black lines denote the ground surface.

**Table 3.** Pulse sequences for Sunds field site.

| Pulse protocol | $\tau$ [ms] | No. periods | Style | $\delta f_T$ [Hz] | Q's | No. stacks |
|---|---|---|---|---|---|---|
| 1 | 10 | 200 | Alt | 0 | 13 | 650 |
| 2 | 10 | 200 | Reg | -5 | 13 | 650 |
| 3 | 10 | 400 | Reg | -2.5 | 13 | 325 |
| 4 | 40 | 800 | Reg | -1.25 | 13 | 162 |

The resutls from Sunds field campaign are shown in Fig. 3a. Multiple boreholes are found in the area all with a water table in the range from 1 m to 7 m. A slight decrease in elevation of the water table is visible towards the north-west part of the area, which is a similar trend as the topography. The SNMR water table estimates are well correlated to most borehole measurements. However, at some locations the water table is estimated deeper than visible from boreholes, especially with S5 and S6. This again could be seasonal changes or that the borehole data has been acquired almost 50 years ago (Borehole ID: 75.853 (GEUS, 2023)).

The resistivity profile in Fig. 3b indicates an upper, approximately 20 m thick resistive unit. Beneath, a more conductive unit is resolved in the southern part of the area with resistivities varying between 10 Ωm and 40 Ωm. The water contents of the resistive unit peak at 25 % to 30 %, and decreases at 10 m to 15 m depths, correlating with the conductive unit in the resistivity profile. The boreholes show a subsurface dominated by a coarse sand, consistent with the observed water content of 25 %. Several deeper (>30 m) boreholes, identify a clay layer at 21 m (Borehole IDs: 75.279, 75.275, 85.842, (GEUS, 2023). This correlates well with the conductive unit from the resistivity measurements and the decrease in mobile water contents. However, water contents were expected to decrease towards a few percent but levels out at above 10 %. When investigating the corresponding $T_2^*$ profiles, the values drop to about $\sim 0.03$ s indicating that the SNMR results have lower sensitivity to the water contents at these depths suggesting very little data influence. As there is little data influence, the data is fitted without altering the water content from the starting model of 10 % in S6. In general, water level estimates from the SNMR in Fig. 3b match well with borehole observations in this unconfined aquifer.

## 3.3 Kompedal field site

The third campaign was completed in Kompedal, a national forest in Central Jutland. Some data regarding the campaign is found in Table 1. The survey is performed in a 23 km$^2$ forest. The area has little infrastructure and yields low noise conditions. A thick sand package deposited by melt water is present in the entire area, yielding a large unconfined aquifer, similar to Sunds. A topographical low is found in the north-west and in south, yet the area is mostly flat. There are limited boreholes in the forest, with more being present in the surrounding area. The boreholes show a water table ranging between 2 and 12 m from north to south in the forest. No frequency offsets are needed with a Larmor frequency of 2155 Hz safely distant from power line noise. Table 4 shows the updated pulse parameters, which yield a combined measurement time of 25 min per site. Two long pulses are added to improve sensitivity at depth.

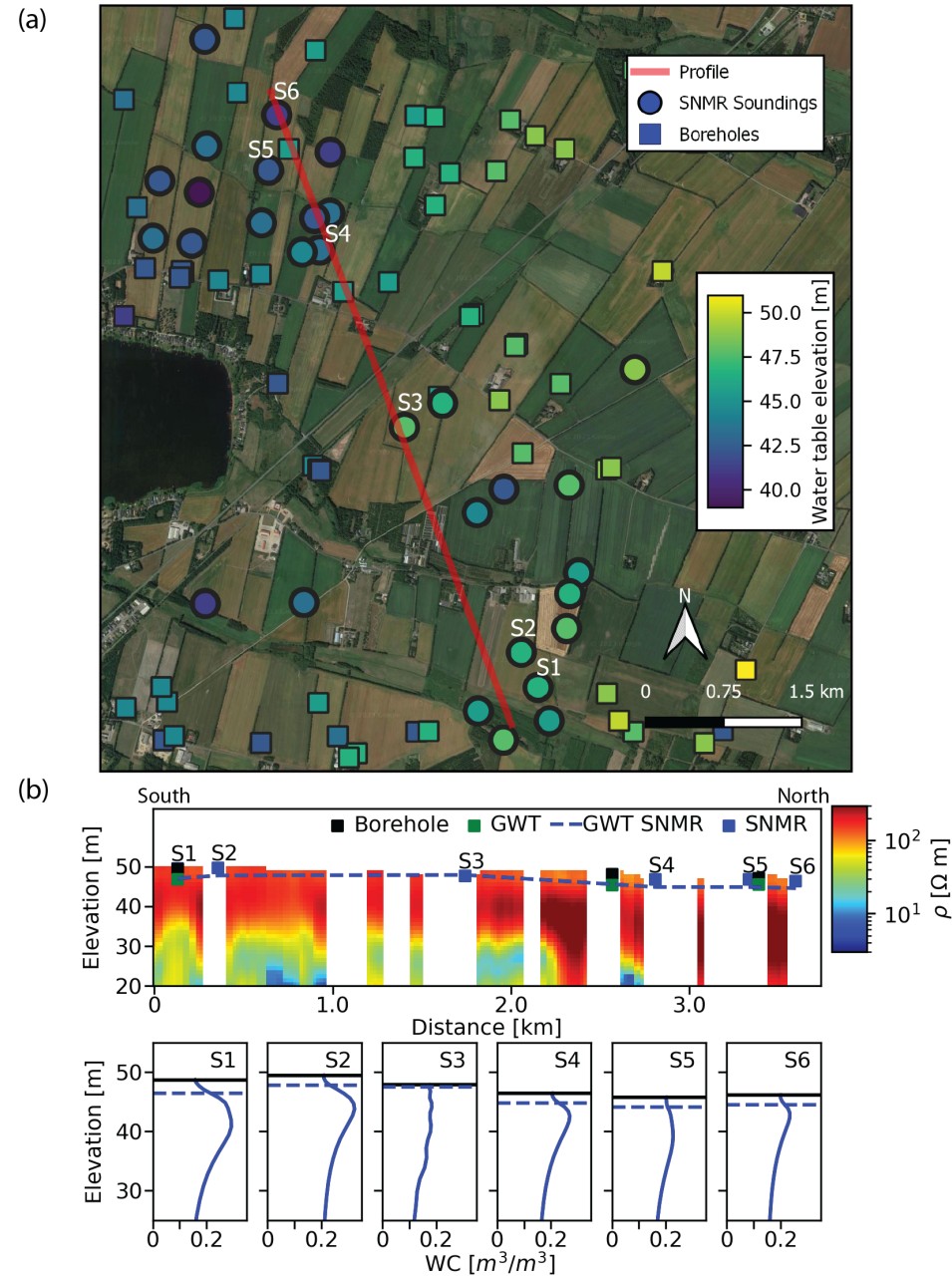

**Figure 3.** (a): Overview image of soundings(circles) and boreholes(squares) from Sunds field site. The color denotes the water table elevation on the same scale for boreholes and SNMR. Profile shows the location of cross-section in (b). Map data: © Google Maps 2021. (b): Cross-section with tTEM resistivities, SNMR water content profiles and with groundwater table (GWT) estimates from SNMR and boreholes. Black lines denote the ground surface.

**Table 4.** Pulse sequences for Kompedal field site.

| Pulse protocol | $\tau$ [ms] | No. periods | Style | $\delta f_T$ [Hz] | Q's | No. stacks |
|---|---|---|---|---|---|---|
| 1 | 10 | 200 | Alt | 0 | 5 | 647 |
| 2 | 20 | 400 | Alt | 0 | 5 | 323 |
| 3 | 40 | 800 | Alt | 0 | 5 | 162 |
| 4 | 60 | 1200 | Alt | 0 | 5 | 108 |
| 5 | 100 | 2000 | Alt | 0 | 5 | 65 |

Figure 4a displays the water level estimates from the Kompedal campaign. Here, the SNMR soundings cover a large area with very limited borehole coverage. The SNMR results show the water table elevation decrease from south-east to north-west, Fig. 4. A rapid change in water table elevation is seen in the south-west. The large water table gradient found in both boreholes and SNMR soundings could indicate a flow direction for groundwater.

The most northern sounding estimates a deep water table compared to nearby boreholes. Water table is estimated at 16 m here, where every layer in the inversion would have a thickness of 2 m. If the water table is estimated at one layer deeper, it would cause a 2 m difference. This is exactly how far the SNMR estimates are from the nearby borehole. Similarly, the most eastern sounding has this effect. The discretization of the model is an important aspect of estimating the layer thicknesses. The discretization reflects the decrease in sensitivity with depth and adding more layers would make the inversion more regularized. As most of the water table depths are 5 m to 10 m these issues are not as profound. The topographical changes are the primary control on the water level with a few deviations.

In Fig. 5a resistivity from a TEM campaign is shown together with SNMR derived results. The resistivity profile reveals an upper resistive unit, interpreted to be the sandy unconfined aquifer. In the northern part, a conductive unit is visible, with resistivities of 10 Ωm to 30 Ωm and an irregular structure, which could be indicative of a conductive layer exposed to glacial tectonics. The water content profiles show a peak water content of approximately 25 %, consistent with borehole observations of sand or gravel deposits (Borehole ID: 76.853, 76.635, 76.631, 76.726 (GEUS, 2023)). The conductive unit coincides with a decrease in water content at 45 m to 55 m elevation for S5, S7, and S8. By the SNMR results alone, the decrease could indicate a unit containing more bound water, i.e., an increase in clay content. The TEM results indicate a conductive unit at these depths, and a borehole north-west identify a till and a Paleogene clay at 20 m to 30 m elevation, respectively (Borehole ID: 76.727 (GEUS, 2023)). Therefore, the layer can be interpreted as a till with less free water than the overlying meltwater sand. Since there is no borehole deeper than 15 m along the profile in Fig. 5a, it is difficult to assert the unit found in TEM and SNMR data as a till.

S6 has a shallow decrease in water content at 55 m to 60 m, which is not aligned with TEM results. When inspecting Fig. 5b, which is perpendicular to Fig. 5a, i.e., S5 and S6 would be at the same position in this figure, the conductive unit appears at 50 m to 60 m, which coincides with the decrease of water content in S6. The projection of SNMR water contents onto the TEM profile is likely the reason for this inconsistency.

Figure 5b shows the second profile visible in Fig. 4. Here, similar structures are visible in the TEM resistivities, with a conductive unit underlying a resistive unit. The SNMR results show water contents peaks of 25 % to 35 %. The peak water content of 35 % in Fig. 5b S13 is located in a small wetland, expected to have very high water content. At S11 in Fig. 5B a decrease in mobile water content is not directly linked to a conductive unit in the TEM results. The TEM profile and SNMR soundings are 100 m to 200 m apart and may thus probe a different subsurface in this glacial landscape. Further, if the change

in geology was from sand to silt, the porosity could have changed, yet resistivity could remain unaltered.

In Fig. 5c another resistivity profile is shown. Here, three boreholes match the SNMR water table estimates. The resistive unit is thicker in this southern part of the area, and the underlying conductive unit is only barely visible underneath. S18 have a different water content profile, which could indicate a lateral change in these meltwater sands and looks like profiles from Sunds in Fig. 3b. The change is not captured by the borehole descriptions (Borehole ID: 76.637-76.641, 76.732, (GEUS,

2023)), which are sparse in detail and only identify meltwater sands and a thin sandy till. The resistivities in the top 30 m are very uniform across the profile, yet the SNMR results at S17 differs from its neighbors. It has a distinct increase in water contents, which indicates a higher porosity more like S18 than S3. By these profiles it is possible to track water table changes on a regional scale and to some extend changes in porosity not captured by either TEM or boreholes.

### 3.4   Data example

A very important feature of these three surveys is the very high data quality. Examples of acquired data are presented in Fig. 6 as sounding curves, i.e., the maximum amplitude of the signal as a function of pulse moment. The figure illustrates representative datasets from each of the campaigns. The error bars are computed as the average noise levels at a set of 8 closely spaced frequencies around the Larmor frequency (-2 Hz to 2 Hz).

The Aars campaign in panel Fig. 6a shows high noise, seen by irregular differences in amplitude between pulse moments

and high error bars. The Sunds campaign, Fig. 6b have low noise at many sites and high signals, due to the very porous shallow saturated aquifer. The Kompedal data set has similar signal amplitudes to Aars and very low noise. The decreased number of pulse moments in 6c yields a rough sounding curve compared to panel b. This feature is an effect of less overlap in spatial sensitivity between each pulse moment. The change in overlapping sensitivity volume between the pulse moments are increased when current stepping is increased (Griffiths et al., 2022). The top 30 m were still resolved by only 5 pulse moments

and 5 different pulses. The high data quality was paramount for this survey and the steady state acquisitions achieved this through 106 soundings in different noise conditions.

### 4   Discussion

A total of 106 soundings, acquired with the novel steady-state approach, are used in water table estimates in this survey, yielding high density maps of the water table. The correlation with independent data such as boreholes highlights the ability

to track water table surfaces in 3D over large areas. Furthermore, comparison with the high spatial coverage of tTEM showed good agreement in finding conductive units as low free water units for the SNMR. The SNMR soundings have been inverted

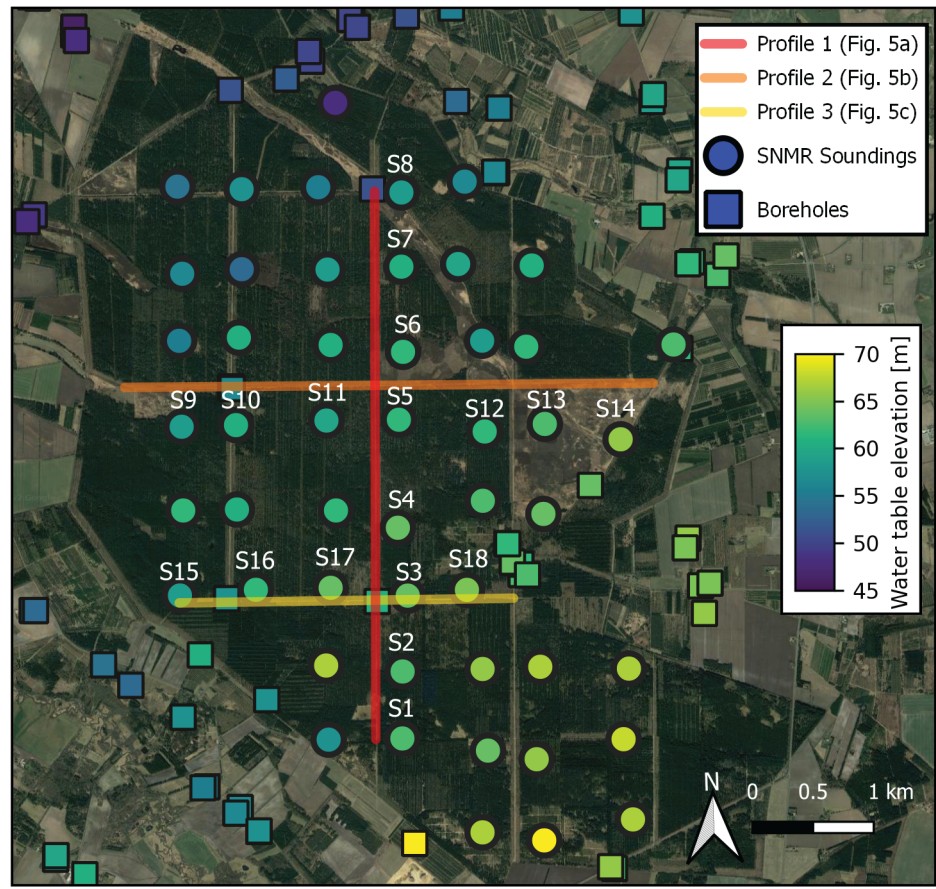

**Figure 4.** Overview of Kompedal field campaign with soundings(circles) and boreholes(squares). The color denotes the water table elevation on the same scale for boreholes and SNMR. The profiles are seen in Fig. 5. Map data: © Google Maps 2021.

as stand-alone models and are not constrained laterally. However, with the data density in this paper, a laterally or spatially constrained inversion could be implemented as seen in previous studies (Behroozmand et al., 2012). This would lead to further data driven consistency between the inversions and will be investigated further in future research.

The SNMR results match well with TEM results at several locations. In some places the SNMR identify low water contents in regions where there is limited to no change in resistivities. Since the measurements are completely independent and their sensitivities are distinct, the methods will resolve different subsurface parameters in some cases. Especially subtle changes in water content will be hard to resolve in the resistivity profiles, as seen in parts of the Kompedal campaign. This highlights the usefulness of a combined approach since SNMR can resolve parameters not easily found by the TEM (Behroozmand et al.,

2012; Irons et al., 2014).

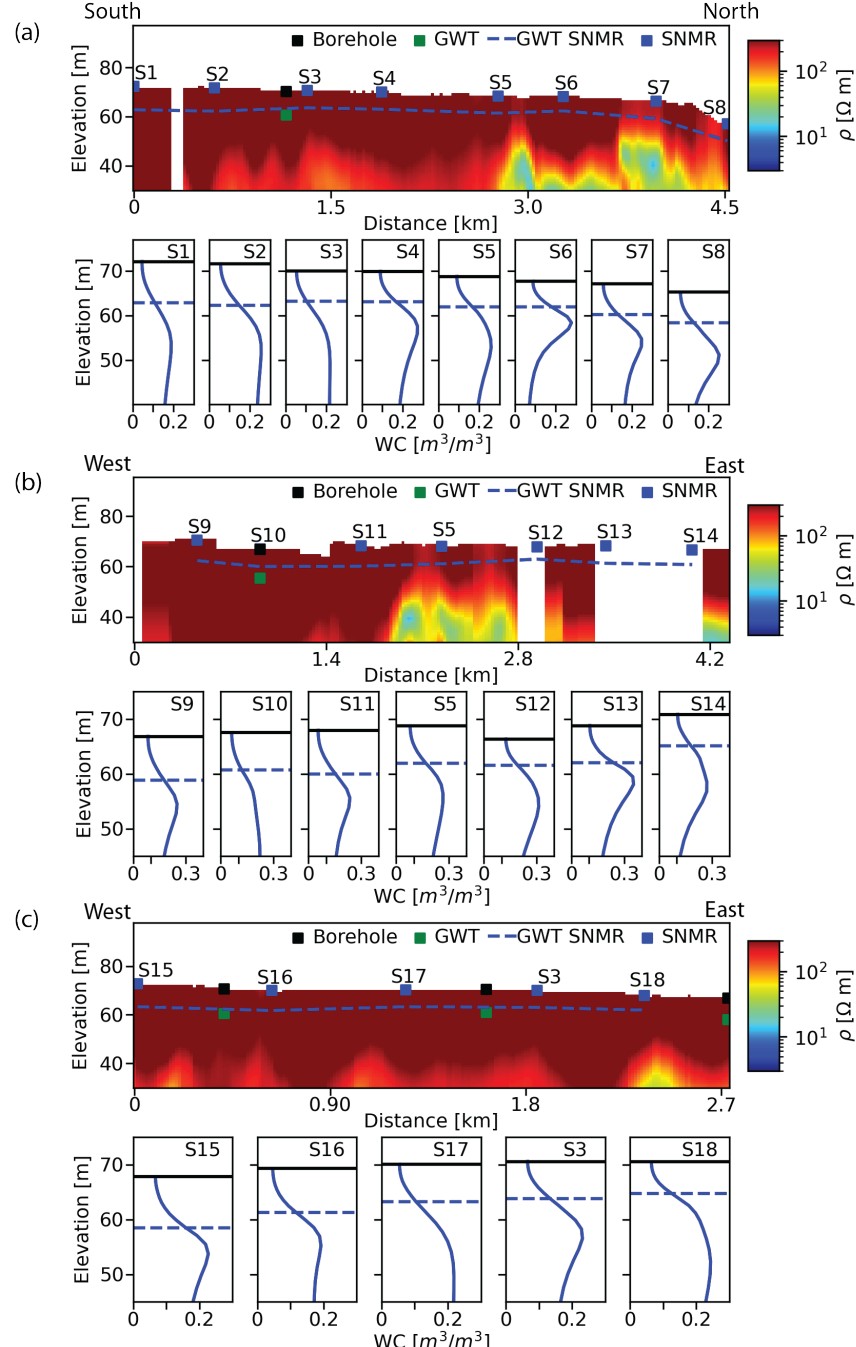

**Figure 5.** (a)-(c): Cross-section with tTEM resistivities, SNMR water content profiles and with groundwater table (GWT) estimates from SNMR and boreholes. Black lines denote the ground surface. The locations are shown in Fig. 4.

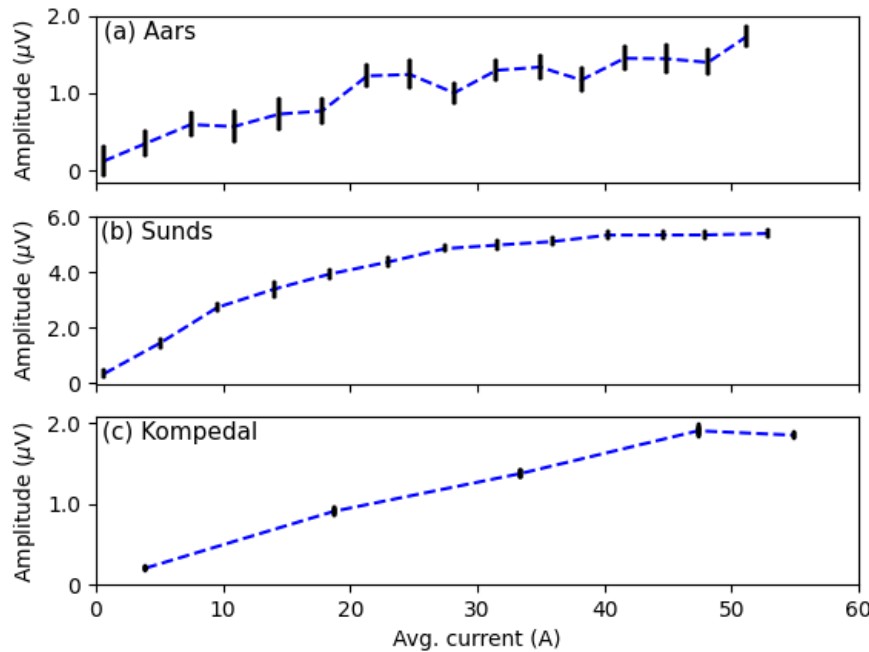

**Figure 6.** Sounding curves of a 10 ms pulse measurement at a) Aars, b) Sunds, and c) Kompedal. Errorbars are shown for each data point based on a 100 Hz band around the signal.

Since the primary results of the study is a regional water table mapping, our smooth inversion scheme is not optimal. A layered inversion is currently not implemented with the steady-state modeling methodology since layer boundaries are fixed using the fast-mapping approach (Griffiths et al., 2022). However, the water level estimation by the largest gradient in the water content profile has been consistent with a majority of borehole measurements. Additionally, the analysis with different
stabilizer functions gave consistent water table estimates.

Through the case studies, pulse sequences have been optimized by heuristically inspecting the data and sensitivity functions. This has led to a decrease in number of pulse moments per site from 64 in Aars to 25 in Kompedal, increasing the number of feasible soundings per day from about two to about ten for a two-person crew, enabled by the rapid acquisition of steady-state data. Further optimization of the field protocols is likely possible and will be considered in future research. By measuring
for only 30 min per site, the temporal variation of the Larmor frequency is limited during the measurements whereas slower approaches are more susceptible to Larmor frequency drift (Legchenko et al., 2016).

The analysis of this study emphasized the water content rather than $T_2^*$ and $T_2$ due to the focus on investigating water table variations. Additional information regarding pore size may be extracted by links to $T_2$ and future research will demonstrate these links with correlation to borehole NMR. The link to hydraulic permeability has been studied and improved in unconsol-

idated material (Dlugosch et al., 2013) and may also be extracted for applications in hydrological modeling. Further research in pulse sequences can lead to more accurate determination and validation of $T_2$.

The aquifers investigated in the study are generally unconfined, where the head measured in wells are equal to where the water resides based on the SNMR measurement. However, this is not the case if an aquifer is confined since the pressure head might exceed the aquitard-aquifer level. Therefore, a comparison between water table estimation of SNMR and borehole measurements is only valid if boreholes are screened in the unconfined aquifers. Another aspect is that the water table estimate from the SNMR includes the capillary fringe. However, the difference would be limited in this study as all aquifers are sand aquifers and would have a small capillary fringe (Bevan et al., 2005) compared to the discretization of the model. Information regarding the aquifer-aquitard boundary in confined aquifers can still be extracted.

The results from this research show the capability of large-scale water table mapping with SNMR using the novel steady-state acquisition. The large-scale mapping enables other applications in hydrological research. The SNMR measurements can be applied to perched aquifers to identify local water tables in these local hydrological systems. Additionally, SNMR data may be implemented in hydrological models to constrain structural settings and inform the model of water bearing units.

## 5   Conclusions

We have used steady-state SNMR to map water table variations over three Danish surveys with over 100 soundings. The fast acquisition of high-quality data enables mapping of large areas with steady state SNMR. Through borehole comparisons, SNMR estimates of regional water table variations are shown to be robust. Furthermore, comparison of SNMR with tTEM data showed consistent structures where conductive clay layers are seen as low mobile water layers in the SNMR results. However, in other places there is not a direct correlation due to the different sensitivities of the two methods. This highlights a usefulness of combining SNMR and TEM for parameter estimation of the subsurface. These techniques can be applied in various hydrological environments and inform on critical parameters for enhanced groundwater knowledge. The iterative improvement of data protocols of the steady state acquisitions made ten sites per day a possibility, thereby enabling large-scale surveys of SNMR. The opportunity to give quantitative estimates of the water level at a regional scale, without well-drilling, as inputs to groundwater models could resolve features previously invisible from borehole data.

*Data availability.* All SNMR and tTEM data in this survey is available at https://doi.org/10.5281/zenodo.8186351.

*Author contributions.* MV acquired, processed, and inverted the data, as well as wrote the article. DG developed the steady-state methodology and helped in interpretations and acquisitions of data. MPG developed the modeling framework and helped in data acquisitions. LL developed the SNMR instrument used in this paper. JJL contributed to writing the article and provided feedback.

*Competing interests.* The author declares that neither they nor their co-authors have any competing interest.

*Acknowledgements.* This work was funded by Independent Research Fund Denmark (9041-00260B).

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
