# Peer review of "Technical note: High density mapping of regional groundwater tables with steady-state surface nuclear magnetic resonance – three Danish case studies"

_Hydrology and Earth System Sciences, 2022_

## Referee Comment (RC1)

Review of

**Technical note: High density mapping of regional groundwater tables with steady-state surface nuclear magnetic resonance – three Danish case studies**

**General comments**

This paper describes the application and results of the steady-state SNMR method recently developed by the Aarhus working group on different sites in Denmark. The water content models of the inversion results (inversion with vertical smoothness constraints) are depicted and compared to transient electromagnetic data. The authors estimate water tables from the maximum gradient of the shallowest water content increase in these models. The water tables are depicted in maps and are compared to the water levels measured in boreholes.

The manuscript is well structured and easy to follow. The measurement progress of the steady-state method compared to the standard SNMR method is impressing. However, I have some concern regarding the relevance of these case studies, at least in the form in which these are presented in this initial version of the manuscript. It reads like a pure documentation of the measurement progress that the new system can make. However, it is not possible to assess the output. Thus, I suggest major revisions:

1. An uncertainty analysis of the water table results is missing. The reader is not able to assess them. Differences between estimated and measured water tables (from boreholes) remain undiscussed. In Fig. 5 discrepancies of several meters can be seen. What is the problem here? Is this an issue for future research? Can it be solved in some future?
2. The authors state that the decrease in the water content models can be attributed to clayey sediments in the subsurface without providing any evidence. They stress that regions with low resistivity (from TEM) indicate clay layers that explain the decrease of the mobile water content, but many water content models are not in agreement with this assumption. Corresponding examples are ignored in the discussion.
3. The sensitivity of the applied pulse sequences with depth should be presented and taken into account for the interpretation of the results. Maybe the decrease of water content mentioned above is simply due to the decreasing sensitivity with depth in many measurements.

In addition to showing at least one example sensitivity function for the SNMR measurements, I further suggest a detailed documentation of at least two data examples – maybe one of the best and one of the worst measurements. This could demonstrate the potential and also the limitation of the proposed SNMR method.

**Specific comments**

P3L83: Fig. 1

P4L90-92: The style of the pulse…
- Please give a reference for the interested reader who wants to learn what the difference is and why it matters to control the polarity of the pulse.

P4L93: The number of pulse moments…
- More explanation is necessary:
  - What is the current range?
  - What impact has the sampling density given by the number of pulses?

P4L95: with -> considering !? Why one minute, not more or less?

P4L96: I suggest switching the last two sentences of this paragraph to refer to the tables in the order in which they actually appear.

P4L97: information regarding -> "general information regarding" or "an overview of…"

P4L101: The spectral analysis approach is not standard in SNMR post-processing, please give a reference.

P5L112: What is a stabilizer function? Please give a reference.

P5L120: It is hard to believe that these alternative regularization approaches will really give "identical" results -> maybe formulations such as "very similar" or "the same with regard to the uncertainty of the measurements" are more appropriate.

P5L129: At what depth does the clay approximately appear? As we learn later, the SNMR results are maybe affected by it.

P6L133, Table 2:
- The label of the first row "pulse" is misleading. Obviously, you apply more than just four pulses. I suggest "pulse protocol" or "scheme".

P6L140: What are the criteria for this heuristic determination? Your statement reads very arbitrarily. It is better to exclude those datasets that do not allow the application of the described procedure.

Even with the data presented (Fig.2), I cannot understand how your procedure can lead to the marked water tables for S6 and S7. For these two, the marked lines do surely not correspond to the maximum gradient of the water content increase. In my optinion it would be better to admit that your procedure cannot be applied to all the data and that future research is necessary on this issue. For S6 and S7, I suggest setting the water tables for S6 and S7 to zero, which seems reliable given the SNMR data.

The same is true for S3 and S5 in Fig.3.

P10L190: This is an effect…
-   Please reformulate this statement. Of course, there will be shallower water tables when the terrain slopes.

P10L200: "Evident" is too strong in this context unless you present ground truth. Again, please provide at least an estimate of the depth at which clay layers were found on your test sites.

P10L204: show_s_

P10L206: Please discuss the decrease of the mobile water content of S3 (Fig.5b) in detail. Here, there are no indications of a conductive layer in the subsurface.

P10L207: Also in Fig.5c, there are undiscussed discrepancies. Regarding the TEM results, the models of S1 to S4 should be very similar. Please discuss why this is not the case.

P10L209: That statement is not true. Please reformulate or erase this sentence. By having all these water table estimates it is possible to track nothing more than the water tables in the region.

P11L217: I totally agree. Do you plan to implement such constraints in the future?

P11-12, Figure 4 and 5: I do not see the point why you focus the analysis on three different profiles in this area, when all these profiles show in principle the same water levels without significant changes. As a matter of fact, there is some variation that could be interesting to focus on, e.g. the two yellow points with water tables at about 15 m. However, these are excluded from the profile analysis. Even if these estimates are not plausible it is much more interesting to discuss them and to learn about the limitations of the method.

Anyway, if you prefer to show different profiles, please label all the different involved measurement points clearly, e.g., from S1 to S20 for the current analysis, and include these labels in the map to guide the reader through these two figures.

P13L247: I cannot accept this conclusion, at least not as direct conclusion of your analysis. Of course, we expect that the content of mobile water descreases with increasing clay content. However, this relationship is not doubtlessly evident by the depicted datasets. For many of the depicted water content models, the decrease actually starts at depths shallower than the corresponding decrease in resistivity, see for instance S1, S2, and S4 in Fig.3 and S6 in Fig.5a. And there are even models where the mobile water content decreases without any indications of having a clay layer in the subsurface, see e.g., S3 in Fig.5b.

---

## Author Response (AR1)

**Manuscript: hess-2022-356**
**Response to reviewers**

Reply: We thank both the reviewer and the editor for their comments to the manuscript. The authors responses will be in blue fonts and the revised manuscript text will be in italics and quotation marks.

**Reviewer 1 comments**

General comments
This paper describes the application and results of the steady-state SNMR method recently developed by the Aarhus working group on different sites in Denmark. The water content models of the inversion results (inversion with vertical smoothness constraints) are depicted and compared to transient electromagnetic data. The authors estimate water tables from the maximum gradient of the shallowest water content increase in these models. The water tables are depicted in maps and are compared to the water levels measured in boreholes.

The manuscript is well structured and easy to follow. The measurement progress of the steady-state method compared to the standard SNMR method is impressing. However, I have some concern regarding the relevance of these case studies, at least in the form in which these are presented in this initial version of the manuscript. It reads like a pure documentation of the measurement progress that the new system can make. However, it is not possible to assess the output. Thus, I suggest major revisions:

Reply: We thank the reviewer for their remarks. The authors have addressed the comments and a detailed explanation is given in the following text.

1. An uncertainty analysis of the water table results is missing. The reader is not able to assess them. Differences between estimated and measured water tables (from boreholes) remain undiscussed. In Fig. 5 discrepancies of several meters can be seen. What is the problem here? Is this an issue for future research? Can it be solved in some future?

Reply: An uncertainty analysis of the water table results is difficult to assess with the deterministic inversion scheme used. We have tested multiple regularization schemes and saw little to no variation. We are using a regularized inversion and are unable to rely on posterior covariance matrix based parameter uncertainties. This will leave only the Bayesian inversion schema which we are currently in the process of implementing. Therefore, an uncertainty analysis is not added in this manuscript. However, the differences between estimated and measured water tables are discussed. Generally, the differences are in the order of a few meters. In the inversion, we are limited by a predefined discretization as stated in the inversion section. This discretization has increasing layer thickness with depth. Therefore, deep estimates such as those found in Fig. 5, is susceptible to defining the ground water table at a layer below or above the true ground water table. For instance, at 10m depth the layer thickness is 1.6m. A finer discretization could enhance the ability to define the water table. Furthermore, the water table estimation from the gradient, is not necessarily the best way of assessing the water table depth but is an unbiased tool to assess the water table from the SNMR results.

P6 L 147-153

*"Some differences between SNMR and boreholes exist, especially in the north part of the area. Here, the SNMR data identify a deeper water table (about x m) than the nearby borehole measurement. This could arise from seasonal changes in water table since the boreholes are done 1.5 year prior to*

*the SNMR soundings. The data can also be explained if the aquifer is confined because the SNMR data is identifying where the water resides rather than the pressure head. The borehole reveals a clay till which could serve as the confining layer (Borehole ID: 40.2055 (GEUS, 2023)).*

P7 L 180-182

The SNMR water table estimates are well correlated to most borehole measurements. However, at some locations the water table is estimated deeper than visible from boreholes, especially with S5 and S6. This again could be seasonal changes or that the borehole data are almost 50 years ago (Borehole ID: 75.853 (GEUS, 2023)

PL218 -222

*"The northern most sounding estimates a deep water table compared to nearby boreholes. Water table is estimated at 16 m here, where every layer in the inversion would have a thickness of 2 m. If the water table is estimated at one layer deeper, it would cause a 2 m difference. This is exactly how far the SNMR estimates are from the nearby borehole. The resolution of the model is an important aspect for determining the water table, and if targets are generally deeper, a different resolution could be used. This is not the case here where most of the water table estimations is ranging from 5 m to 10 m."*

2. The authors state that the decrease in the water content models can be attributed to clayey sediments in the subsurface without providing any evidence. They stress that regions with low resistivity (from TEM) indicate clay layers that explain the decrease of the mobile water content, but many water content models are not in agreement with this assumption. Corresponding examples are ignored in the discussion.

Reply: We thank the reviewer for this comment. For each of the campaigns, a more nuanced discussion of the discrepancies encountered is added. Especially in Fig. 5 discrepancies are seen where the SNMR profiles are projected 200m to 300m onto the TEM resistivity profiles. This is because TEM and SNMR data comparisons are not formed of spatially overlapping data (but nearby). This could account for some of the inconsistencies since geology could vary within the 200m from SNMR to TEM measurements, but a discussion on these is added to highlight various explanations. Borehole IDs are added to statements of geology with a reference to the online free database where all boreholes are present.

P9 L191-192

*"This duality of $T*2$ and water contents could explain some of these consistencies and a limitation of only using the water contents as a mean of identifying aquitards."*

P11 L224-227

*"S6 has a shallow decrease in water content at 55 m to 60 m, which is not aligned with TEM results. When inspecting Fig. 5b, which is perpendicular to Fig. 5a, i.e., S5 and S6 would be at the same position in this figure, the conductive unit appears at 50 m to 60 m, which coincides with the decrease of water content in S6. Therefore, most of the discrepancy at this location is likely due to how the SNMR and TEM results being placed several hundreds of meters away from each other."*

P11 L230-236

*"At S11 in Fig. 5B a decrease in mobile water content is not directly linked to a conductive unit in the TEM results. Since the profile is located along a TEM line, the SNMR water content profiles are projected about 100 m to 200 m onto the resistivity profile. The geology changes quite rapidly in*

*these glacial landscapes, which could explain parts of these differences. Additionally, TEM and SNMR arises from different geophysical phenomena, which implies that a change in water content is not necessarily seen in the resistivity profile and vice versa. If the change were from sand to silt, the porosity could have changed, yet resistivity could remain unaltered."*

3. The sensitivity of the applied pulse sequences with depth should be presented and taken into account for the interpretation of the results. Maybe the decrease of water content mentioned above is simply due to the decreasing sensitivity with depth in many measurements.

Reply: Griffiths et al., 2022 presents sensitivities of similar pulses with similar current ranges. The short pulses, i.e., 10ms and 20ms pulses, are not very sensitive below 20m depth. However, the 40ms pulse sensitivity peaks around 25m at high currents and this pulse is used in all 3 campaigns. For this reason, we are capable of interpreting on data from the first 30m with confidence. We will add details on the depth of investigation to the revised manuscript. A more rigorous estimation of the depth of investigation is the subject of on-going parallel research.

P4 L100-103
*"The pulses are sampled with the same current range, 5A to 80A, which yield some overlapping sensitivities. The pulse protocols and current amplitudes were chosen based on having relaxation time and spatial sensitivity. The depth of investigation is about 30m for the soundings constrained by the coil size and maximum current."*

In addition to showing at least one example sensitivity function for the SNMR measurements, I further suggest a detailed documentation of at least two data examples – maybe one of the best and one of the worst measurements. This could demonstrate the potential and also the limitation of the proposed SNMR method.

Reply: Thank you for your comment. A figure containing data from each of the campaigns has been added. The sites are picked at random and generally show the trends and noise found at the different areas of interest. These sites are representative of the majority of data within each survey. To show one of the worst measurements would not be valuable since these have been removed from the water table estimates due to high noise conditions.

P12 L245-257

*"Data example*

*A very important feature of these three surveys is the very high data quality. Examples of acquired data are presented in Fig. 6 as sounding curves, i.e., the maximum amplitude of the signal as a function of pulse moment. The figure illustrates representative datasets from each of the campaigns. The error bars are computed as the average noise levels at a set of 8 closely spaced frequencies around the Larmor frequency (-2 Hz to 2 Hz).*
*The Aars campaign in panel Fig. 6a shows high noise, seen by irregular differences in amplitude between pulse moments and high error bars. The Sunds campaign, Fig. 6b have low noise at many sites and high signals, due to the very porous shallow saturated aquifer. The Kompedal data set has similar signal amplitudes to Aars and very low noise. The decrease amount of pulse moments in 6c yields a rough sounding curve compared to panel b. This feature is an effect of less overlap in spatial sensitivity between each pulse moment. The change in overlapping sensitivity volume between the pulse moments are increased when current stepping is increased (Griffiths et al., 2022). The top 30 m were still resolved by only 5 pulse moments and 5 different pulses. The high data quality was*

*paramount for this survey and the steady state acquisitions achieved this through 106 soundings in different noise conditions."*

P3L83: Fig. 1

Reply: Typo has been fixed in revised manuscript.

P4L90-92: The style of the pulse…

- Please give a reference for the interested reader who wants to learn what the difference is and why it matters to control the polarity of the pulse.

Reply: A reference is added to the style description for the already cited:
Grombacher, D., Griffiths, M., Liu, L., Vang, M., and Larsen, J.: Frequency Shifting Steady-State Surface NMR Signals to Avoid Problematic Narrowband-Noise Sources, Geophysical Research Letters, 49, e2021GL097 402, https://doi.org/10.1029/2021GL097402, 2022.

P4L91-92

*"The style of the pulse can be either "regular", where the polarity of the pulses is the same for the full pulse train, or "alternating" where the polarity changes for each pulse (Grombacher et al., 2022)."*

P4L93: The number of pulse moments…

- More explanation is necessary:

o What is the current range?

o What impact has the sampling density given by the number of pulses?

Reply: The current range is from 5-80 A and is sampled linearly. Added description of the current range at P4L93-96:

*"The number of pulse moments (Q ̓ s) defines how dense the current range is sampled. In this study, a current range from 5 A to 80 A was acquired for high resolution of the shallow subsurface. By increasing the current amplitude, the spatial sensitivity is changed from shallow to deep as in regular SNMR. Furthermore, current drooping is encountered over 80 A, which limits the maximum current."*

The current density could possibly be decrease by having 4 different pulses with much overlapping sensitivity, seen in Griffiths et al. (2022). It explains how we are able to decrease measurement time between campaigns ending up at 25 min of acquisition time per site.

The employed current densities decrease between the surveys due to our improving field practices, where data redundancy in collected data suites has been continually reduced. Future research will focus on minimizing the acceptable current densities that still deliver satisfactory resolution.

P4L95: with -> considering !? Why one minute, not more or less?

Reply: Changed "with" to "considering".

One minute was chosen based on field observations that it generally returns a high signal to noise ratio for Aars. One minute is simply a convenient duration in that it allows quick back of the envelope calculation of the total measurement duration at each site. For comparison between campaigns, the same stacking time were used for the other two campaigns. If noise conditions

require it, the stacking time can easily be increased, but this will decrease the number of soundings per day.

P4 L97-99

*"In this study, the number of stacks is chosen based on 1 minute of acquisition considering the associated repetition time. The 1-minute acquisition asserted a high data quality in Aars while maintaining a short measurement time at each site and was preserved for the following campaigns for easy comparison."*

P4L96: I suggest switching the last two sentences of this paragraph to refer to the tables in the order in which they actually appear.

Reply: We thank the reviewer for this comment. The order of the sentences has been switched in the revised manuscript.

P4 L104-106

*"In Table 1 general information regarding the campaigns is shown. Pulse sequences are updated between campaigns in particular, a decrease in the current sampling increases the acquisition rate while maintaining high-quality data. Tables 2, 3, and 4 describes the pulse sequences for each of the campaigns."*

P4L97: information regarding -> "general information regarding" or "an overview of…"

Reply: Sentence changed to "*In table 1 general information regarding the campaigns is shown.*"

P4L101: The spectral analysis approach is not standard in SNMR post-processing, please give a reference.

Reply: A reference is added which describes the spectral analysis approach:
Liu L, Grombacher D, Auken E, Larsen JJ. Complex envelope retrieval for surface nuclear magnetic resonance data using spectral analysis. Geophysical Journal International. 2019 May;217(2):894-905.

P5 L108-109

*"Furthermore, a spectral analysis approach based on the discrete Fourier transform is used to retrieve the NMR signal from the time series (Liu et al., 2019)."*

P5L112: What is a stabilizer function? Please give a reference.

Reply: Reference added to this line.

P5 L120-121

*"A stabiliser function is used in the inversion to ensure convergence (Grombacher et al., 2017)."*

P5L120: It is hard to believe that these alternative regularization approaches will really give "identical" results -> maybe formulations such as "very similar" or "the same with regard to the uncertainty of the measurements" are more appropriate.

Reply: "Identical" has been changed to "very similar" in the revised manuscript.

P5L129: At what depth does the clay approximately appear? As we learn later, the SNMR results are maybe affected by it.

Reply: Specific depths and borehole IDs has been added to the manuscript to refer to the evidence of clayey sediments. The depth of the clay encountered by boreholes are 50m but the TEM results reveal that the layer is sloping and occurs at around 10m East in the area of interest.

P6 L137-138

*"A Paleogene clay is underlying the glacial deposits located at depths of 60 m to 50 m (Borehole ID: 40.1006, 48.1171 (GEUS, 2023))."*

P6L133, Table 2:

- The label of the first row "pulse" is misleading. Obviously, you apply more than just four pulses. I suggest "pulse protocol" or "scheme".

Reply: The table's first row will now read as "*Pulse protocol*". The change has been applied for each campaign table.

P6L140: What are the criteria for this heuristic determination? Your statement reads very arbitrarily. It is better to exclude those datasets that do not allow the application of the described procedure.

Reply: By inspection, the relaxation parameters showed a clear transition which were used in determining the water table in an otherwise limited structure in the water content profile. The two water table estimates have been omitted from the figure.

Even with the data presented (Fig.2), I cannot understand how your procedure can lead to the marked water tables for S6 and S7. For these two, the marked lines do surely not correspond to the maximum gradient of the water content increase. In my optinion it would be better to admit that your procedure cannot be applied to all the data and that future research is necessary on this issue. For S6 and S7, I suggest setting the water tables for S6 and S7 to zero, which seems reliable given the SNMR data.

Reply: For S7 there is a small decrease in water contents followed by a slight increase which is found to have the highest positive gradient. However, we agree that the water tables could be set to zero as to where the maximum water content is encountered, and still be used in the final water table map, without exclusion of the datasets based on the previous comment. Based on feedback from the second reviewer, we are now using the elevation of the water table instead of water table depth, with the elevation of the water table set to the topography, i.e., water table depth equal to zero.

P6 L153-154

*"At S6 and S7, high water contents were found close to the surface and the water table is set to the topography in these soundings."*

The same is true for S3 and S5 in Fig.3.

Reply: The water tables of S3 and S5 in Fig. 3 has been changed to topography level.

P10L190: This is an effect…

- Please reformulate this statement. Of course, there will be shallower water tables when the terrain slopes.

Reply: The sentence has been deleted. Since topography is included in the maps, this formulation is no longer necessary.

P10L200: "Evident" is too strong in this context unless you present ground truth. Again, please provide at least an estimate of the depth at which clay layers were found on your test sites.

Reply: A less loaded word has been added with a description of where the clay is located by the TEM and a description of where the clay layers are intercepted is now given.

*"By the SNMR results alone, the decrease could indicate a less saturated unit or a unit containing more bound water. The TEM results indicate a conductive unit at these depths, and a borehole north-west perturb a till and a Paleogene clay at 20 m to 30 m elevation, respectively (Borehole ID: 76.727 (GEUS, 2023)). Therefore, the layer can be interpreted as a till with less free water than the overlying meltwater sand. Since there is no borehole deeper than 15 m along the profile in Fig. 5a, it is difficult to assert the unit found in TEM and SNMR data as a till."*

P10L204: shows

Reply: Fixed

P10L206: Please discuss the decrease of the mobile water content of S3 (Fig.5b) in detail. Here, there are no indications of a conductive layer in the subsurface.

Reply: Thank you for making us aware of this. Unfortunately, the wrong water content profile was set at this location (from a more northern sounding, where the conductive unit is visible) and for S6 in the same profile. We apologize for the inconvenience, and we have fixed the issue while checking all other soundings, which were right. The decrease is not as profound yet is still present and will be discussed in the text. We have also added a description on how different sensitivities for the methods yield these differences.

P11 L230-236

*"At S11 in Fig. 5B a decrease in mobile water content is not directly linked to a conductive unit in the TEM results. Since the profile is located along a TEM line, the SNMR water content profiles are projected about 100 m to 200 m onto the resistivity profile. The geology changes quite rapidly in these glacial landscapes, which could explain parts of these differences. Additionally, TEM and SNMR arises from different geophysical phenomena, which implies that a change in water content is not necessarily seen in the resistivity profile and vice versa. If the change were from sand to silt, the porosity could have changed, yet resistivity could remain unaltered."*

P10L207: Also in Fig.5c, there are undiscussed discrepancies. Regarding the TEM results, the models of S1 to S4 should be very similar. Please discuss why this is not the case.

Reply: Differences between TEM and SNMR results are discussed in the revised manuscript. One possible explanation is that we are projecting the SNMR measurements on to the TEM resistivities 200m away. Further, a change in water content can occur without a severe affect on the resistivity measured by the TEM method. The outlier of these is S3 where a decrease in water contents is not visible after the peak. Further it seems that at S3 a higher maximum water content is found here. This could indicate a higher porosity, or a coarser material. The effect on TEM results, would be very limited, since the sand is quite resistive in profile.

P11 L238-243

*"S18 have a different water content profile, which could indicate a lateral change in these meltwater sands and looks like profiles from Sunds in Fig. 3b. The change is not captured by the borehole descriptions (Borehole ID: 76.637-76.641, 76.732, (GEUS, 2023)), which are sparse in detail and only*

*identify meltwater sands and a thin sandy till. The resistivities in the top 30 m are very uniform across the profile, yet the SNMR results at S17 differs from its neighbors. It has a distinct increase in water contents, which indicates a higher porosity more like S18 than S3."*

P10L209: That statement is not true. Please reformulate or erase this sentence. By having all these water table estimates it is possible to track nothing more than the water tables in the region.

Reply: The statement has been reformulated to a clearer statement that it is possible to track the water table in the region.

P11 L243-244

*"By these profiles it is possible to track water table changes on a regional scale and to some extend changes in porosity not captured by either TEM or boreholes."*

P11L217: I totally agree. Do you plan to implement such constraints in the future?

Reply: Yes, it is in the pipeline to implement LCI and possibly SCI for the SNMR inversions. Preliminary results show great promise for constraining relaxation parameters using LCI and future work will be focused on this topic.

P11-12, Figure 4 and 5: I do not see the point why you focus the analysis on three different profiles in this area, when all these profiles show in principle the same water levels without significant changes. As a matter of fact, there is some variation that could be interesting to focus on, e.g. the two yellow points with water tables at about 15 m. However, these are excluded from the profile analysis. Even if these estimates are not plausible it is much more interesting to discuss them and to learn about the limitations of the method.

Reply: The three profiles were based on close TEM measurements, since acquisition of this data was limited to gravel roads between the trees. Further, after reviewing comments from the second reviewer, the water table depths figures were changed to indicate water table elevation, which diminished the outliers greatly. Therefore, the authors have decided not to change the profiles in the plots.

Anyway, if you prefer to show different profiles, please label all the different involved measurement points clearly, e.g., from S1 to S20 for the current analysis, and include these labels in the map to guide the reader through these two figures.

Reply: Each sounding has been numbered from S1 to S18 since two of the soundings occur twice in the profiles. Additionally, topography has been added to the profiles while all axes are now elevations instead of depth below surface.

P13L247: I cannot accept this conclusion, at least not as direct conclusion of your analysis. Of course, we expect that the content of mobile water descreases with increasing clay content. However, this relationship is not doubtlessly evident by the depicted datasets. For many of the depicted water content models, the decrease actually starts at depths shallower than the corresponding decrease in resistivity, see for instance S1, S2, and S4 in Fig.3 and S6 in Fig.5a. And there are even models where the mobile water content decreases without any indications of having a clay layer in the subsurface, see e.g., S3 in Fig.5b.

Reply: The conclusion will be changed to a more varied conclusion mentioning the different sensitivities of the two methods and how they can add knowledge in regard to resistivities and water

contents, respectively. A description of the differences between TEM and SNMR results have been added in the results and discussed further.

P14 L301-302

*However, in other places there is not a direct correlation due to the different sensitivities of the two methods. This highlights a usefulness of combining SNMR and TEM for parameter estimation of the subsurface.*

**Reviewer 2 comments**

General

The team around Aarhus University and Aarhus Centre for Water Technology have further developed the technique of Surface Nuclear Magnetic Resonance and have introduced the approach of steady state pulse sequences. In the near past they have published a series of scientific articles on this topic where they systematically investigate different aspects of the fundamentals, measurements, and processing and inversion. The present manuscript describes case studies that are based on the previous work.

The present "technical note" presents data from such measurements at three different sites and bring them in context with complementary information from boreholes and 2D-profiles of electromagnetic measurements.

In general, the manuscript provides data of quality, relevant information for the scientific community and a good standard of scientific conduct.

Reply: We thank the reviewer for the comments, and we have addressed the shortcomings in the revised manuscript.

**Structure and Content**

Even though the shown data is of good quality and relevance, the manuscript is lacking clear statements. For example:

The application of a new measurement device and the new sounding approach. Is that relevant for the paper? Do the authors want to present the advantage of SS-NMR over FID-NMR. In this case the presented data and corresponding discussion do not reveal this.

Reply: Our focus is rapid mapping enabled by this approach, with the scope being the ability to map large areas and not a direct comparison between FID and steady-state acquisitions.
For a direct comparison between FID-NMR and steady state NMR, see Grombacher et al, (2021) where a detailed section illustrates the advantage.

Correlation with local and regional groundwater regimes. For each site piezometric data from nearby boreholes are presented. Yet, the message is not clear. Is NMR complementary data to piezometric data or does it replace these? Or is it the combination like for confined/artesic aquifers as explained in the text but not supported by data.

Reply: We thank the reviewer for this comment. We have added details to the manuscript to highlight that these SNMR soundings are independent measurements. The SNMR results can be used independently in unconfined aquifers to draw conclusions on flow base on head gradient without

having boreholes as validation. In this manuscript we compare SNMR soundings with water table measurements from piezometers to validate our use of the steady state acquisition for large scale groundwater mapping. Further, it illustrates that the SNMR can be used as a complement to piezometers or borehole water level measures, as an infill between boreholes. However, it is noted that this is not the case in confined/artesian aquifers as the reviewer mentions, since the SNMR is mapping where water resides and not the actual pressure head which determines flow direction.

P13 L266-270

*"The SNMR results match well with TEM results at several locations. In some places the SNMR identify low water contents in regions where there is limited to no change in resistivities. Since the measurements are completely independent and their sensitivities are distinct, the methods will resolve different subsurface parameters in some cases. Especially subtle changes in water content will be hard to resolve in the resistivity profiles, as seen in parts of the Kompedal campaign. This highlights the usefulness of a combined approach since SNMR can resolve parameters not easily found by the TEM."*

The relevance of the TEM resistivity sections is also not obvious. Water contents and resistivities correlate in different fashion for the different sites and partially no correlation is visible or even contradictive. What exactly is the intention of the authors to demonstrate?

Reply: The TEM resistivity sections were shown to compare adjacent geophysical measurements in the regions with sparse borehole coverage. Further, in areas dominated by meltwater sands and conductive tills, the TEM and NMR signals can be interpreted together as a drop in free water below water table, could coincide with a conductive structure identifying clay structures. Another reason for the comparison is to establish that resistivity measurements are not always capable of resolving water tables, showcasing the capabilities of a combined mapping of SNMR and TEM. The resistivities and water contents correlate in some areas while in others it is hard to determine a trend. In general, the SNMR and TEM has different sensitivities and do not always have comparable results. Yet, in some cases if a trend is visible in both methods, it can be used to resolve a given subsurface layer's properties.
We have added more explanation and discussion of differences between SNMR and TEM measurements in the revised manuscript.

P7 L180-181

*"However, at some locations the water table is estimated deeper than visible from boreholes, especially with S5 and S6. This again could be seasonal changes or that the borehole data are almost 50 years ago (Borehole ID: 75.853 (GEUS, 2023))"*

P11 L220-227

*"The TEM results indicate a conductive unit at these depths, and a borehole north-west perturb a till and a Paleogene clay at 20 m to 30 m elevation, respectively (Borehole ID: 76.727 (GEUS, 2023)). Therefore, the layer can be interpreted as a till with less free water than the overlying meltwater sand. Since there is no borehole deeper than 15 m along the profile in Fig. 5a, it is difficult to assert the unit found in TEM and SNMR data as a till. S6 has a shallow decrease in water content at 55 m to 60 m, which is not aligned with TEM results. When inspecting Fig. 5b, which is perpendicular to Fig. 5a, i.e., S5 and S6 would be at the same position in this figure, the conductive unit appears at 50 m to 60 m, which coincides with the decrease of water content in S6. Therefore, most of the discrepancy at this location is likely due to how the SNMR and TEM results being placed several hundreds of meters away from each other."*

P11 L238-244

*"S18 have a different water content profile, which could indicate a lateral change in these meltwater sands and looks like profiles from Sunds in Fig. 3b. The change is not captured by the borehole descriptions (Borehole ID: 76.637-76.641, 76.732, (GEUS, 2023)), which are sparse in detail and only identify meltwater sands and a thin sandy till. The resistivities in the top 30 m are very uniform across the profile, yet the SNMR results at S17 differs from its neighbors. It has a distinct increase in water contents, which indicates a higher porosity more like S18 than S3."*

P13 L266-270

*"The SNMR results match well with TEM results at several locations. In some places the SNMR identify low water contents in regions where there is limited to no change in resistivities. Since the measurements are completely independent and their sensitivities are distinct, the methods will resolve different subsurface parameters in some cases. Especially subtle changes in water content will be hard to resolve in the resistivity profiles, as seen in parts of the Kompedal campaign. This highlights the usefulness of a combined approach since SNMR can resolve parameters not easily found by the TEM.*

NMR

The presentation of real NMR surveys with the novel approach is relevant and the data is in general useful for this demonstration. Yet, the authors miss to outline some relevant statements. The aspects of i) frequency offset, ii) tau, iii) pulse moments, iv) regular/alternating pulses, are addressed in section 2.3. Yet, from the description of the data collection at the different sites it is not clear which of the aspects is key to the success of the measurements. Is there anything special in choosing or varying these parameters or is their choice arbitrary?

Reply: It is true that the steady-state acquisitions introduce a number of additional experimental parameters into the data suite – beyond the typical pulse length and current amplitudes. The data presented in this work span first year of development and contains changes in field protocols. The reason our field protocols collect with multiple tau, offsets, and current amplitude is to balance spatial and relaxation time sensitivity. Griffiths et al., (2022) show examples of sensitivity which illustrate how sensitivity varies with currents and different taus. But a more comprehensive look at how each of these parameters affect the depth of the signal is the focus of on-going work. The frequency offset is discussed in greater detail in Grombacher et al., 2022.

The field protocols have been incremented over time to reduce the total number of measurements, but while preserving an ability to produce a satisfactory resolution image. The combinations employed were selected based on examination of sensitivity kernels, and an attempt to remove data redundancy between measurements that shared heavily overlapping spatial/relaxation sensitivities.

Regarding the frequency offset and alternating pulses, the decision in Sunds was made to minimize the influence of a co-frequency harmonic issue. Sunds had a local co-frequency harmonic issue requiring offset, which is described in more detail in Grombacher et al, 2022. The frequency offsets were not necessary for the other campaigns since the Larmor Frequency was several hertz offset from the powerline harmonic.

We have changed parts of the Data collection section to further explain how we have chosen these pulse parameters for each campaign.

P4 L86-103

*"The number of pulse moments (Q's) defines how dense the current range is sampled. In this study, a current range from 5 A to 80 A was acquired for high resolution of the shallow subsurface. By increasing the current amplitude, the spatial sensitivity is changed from shallow to deep as in regular SNMR. Furthermore, current drooping is encountered over 80 A, which limits the maximum current. The number of stacks is how many times each measurement is repeated with the same pulse moment. In this study, the number of stacks is chosen based on 1 minute of acquisition with considering the associated repetition time. The 1-minute acquisition asserted a high data quality in Aars while maintaining a short measurement time at each site and was preserved for the following campaigns for easy comparison. Multiple pulses with varying pulse parameters are used to fully resolve the subsurface. The pulses are sampled with the same current range, 5 A to 80 A, which yield some overlapping sensitivities. The pulse protocols and current amplitudes were chosen based on having relaxation time and spatial sensitivity. "*

One important information remains largely unclear to me in the application of the novel SS-SNMR approach compared to conventional FID-SNMR. In FID-SNMR the spatial sensitivity is varied be increasing the pulse-moment. This results from the effect that successively deeper parts of the subsurface are excited by ~90° angles of excitation while shallower parts mutually cancel out at multiple revolutions. In this manuscript and the cited literature, I did not find a conclusive explanation how the spatial sensitivity varies for SS-SNMR and how then the water content profile with depth is inverted. I apologize if I have missed this information, the authors were quite productive in publishing papers on this topic and the review of the present manuscript required a bit of reading supporting material. The cited article of Griffith 2022 is still not published and was not available for this review.

Reply: The steady state NMR signal's depth is controlled in a similar manner as FID measurements. A sounding approach that increases currents excites deeper parts of the subsurface. This allows both the pulse duration and current amplitude to be manipulated to vary the signal's depth. However, for steady-state measurements several additional factors can now also influence the depth, including the repetition times, alternating versus regular, and the relaxation times at depth. But in practice, the data suites employed (that contain multiple of each of these parameters) provide measurements that all have differences in spatial sensitivity – taken together they still deliver the required information needed to produce a depth profile.

A more comprehensive examination of factors controlling the steady-state signal's depth is the subject of on-going research.

For the Aars field site in Table 2 the authors list 4 different tau at 16 Q each. In Table 1, they list 64 as no. of Q's. I conclude that the total number of Qs is the number of different currents times the number of different tau, is that correct? (Does not apply for the Sound site). And do the different resulting Ernst angles lead to different spatial sensitivities? This needs to be explained or referenced in more detail.

Reply: The 64 measurements are not directly pulse moments, but rather different currents with different pulse protocols. They are not different pulse moments since the 10ms alternating and 10ms regular pulses would have identical current amplitudes, thereby an identical pulse moment. But as seen in Griffiths et al., (2022) Fig. 6, the pulses still have different spatial sensitivity.

We wouldn't think of the different sequences having different Ernst angles, and therefore having different spatial sensitivities. Rather that the different sequences have different performance across the full range of B1 present at depth, and therefore have different spatial sensitivities. The important difference being that the sequences are different in every voxel in the subsurface – not only the specific voxel where the Ernst angle perturbation is occurring.
A more detailed description of the pulse protocols is added to the revised manuscript and now reads:

P4 L100-103

*"Multiple pulses with varying pulse parameters are used to fully resolve the subsurface. The pulses are sampled with the same current range, 5 A to 80 A, which yield some overlapping sensitivities. The pulse protocols and current amplitudes were chosen based on having relaxation time and spatial sensitivity. The depth of investigation is about 30 m for the soundings constrained by the coil size and maximum current."*

Presentation of field results

A major shortcoming of the manuscript is the compilation of the figures. The authors mix between "Elevation" (above sea level?) for the resistivity cross sections and groundwater tables, and "depth below surface" for the single SNR soundings and the maps. This is largely confusing. In the maps the "depth below surface" is shown for NMR derived water tables and boreholes, but no information about topography is given (only partially qualitatively in the text). The major information about the applicability of SNMR for local groundwater table estimates is therefore missing.

Reply: We thank the reviewer for this comment. After revising the figures, we now only show elevations of water table and the SNMR water content profiles are converted to elevation, with ground surface indicated in every subfigure. After correction, SNMR show even greater resemblance with borehole measurements of water table. Now it is possible to assess groundwater level patterns and possibly flow paths by head gradient.

In various cases, the authors refer to a good agreement of the water content profiles and resistivity sections with the geology observed in the boreholes. But no geologic profiles from boreholes is shown.

Reply: Instead of plotting the borehole data, borehole IDs has been added with a reference to the online open-source Jupiter database (the Danish National Borehole Database operated by the Geological Survey of Denmark and Greenland). Here, full borehole descriptions can be viewed for all areas. We believe that this gives the reviewer or possible readers the ability to see all data used in the comparisons. Examples of these in the revised manuscript:

P6L 137-138

*"A Paleogene clay is underlying the glacial deposits located at depths of 60 m to 50 m (Borehole ID: 40.1006, 48.1171 (GEUS, 2023))."*

P6 L151-152

*"The borehole reveals a clay till which could serve as the confining layer (Borehole ID: 40.2055 (GEUS, 2023))."*

P7L 180-181

*"This again could be seasonal changes or that the borehole data are almost 50 years ago (Borehole ID: 75.853 (GEUS, 2023))."*

P11 L217-218

*"The water content profiles show a peak water content of approximately 25 %, consistent with borehole observations of sand or gravel deposits (Borehole ID: 76.853, 76.635, 76.631, 76.726 (GEUS, 2023))."*

P11 L220-222

*"The TEM results indicate a conductive unit at these depths, and a borehole north-west perturb a till and a Paleogene clay at 20 m to 30 m elevation, respectively (Borehole ID: 76.727 (GEUS, 2023))."*

P11 L240-241

*"The change is not captured by the borehole descriptions (Borehole ID: 76.637-76.641, 76.732, (GEUS, 2023)), which are sparse in detail and only identify meltwater sands and a thin sandy till."*

Conclusion

The information presented in the present manuscript is very relevant to the scientific and technical community. Yet, the scope of the study is not well presented. With moderate effort to rearrange parts of the text, sharpen the scope of the study and improve the figures, the manuscript can be brought to an acceptable standard.

---

## Referee Report (RR1)

**Technical note: High density mapping of regional groundwater tables with steady-state surface nuclear magnetic resonance – three Danish case studies**

**General assessment**

The authors touched many aspects of the first review in their revision. By consequently showing the elevation of water tables instead of depth below surface as in the first version, the estimates from the SNMR data can be assessed as being much more plausible than before. The discussion of the results has been extended by introducing possible reasons for the remaining discrepancies between the different data sets.

However, there are still inconsistencies that need to be clarified, and, unfortunately, the manuscript still has significant linguistic deficits that have to be corrected before publication. Thus, I suggest moderate revisions.

**Details**

The page and Line numbers refer to the document with marked changes (hess-2022-356-ATC1_comments.pdf)

P4L96: The sentence "Furthermore,…" can be erased or should be reformulated.

P4L97: with > at

P4L102: redundant information

P4L103: What do you mean by "having relaxation time and spatial overlapping"? A tradeoff between resolving spatial information and discovering the relaxation times of the signals? Please clarify.

P4L101-103: As already suggested in my first review, I would prefer a figure with an example of a sensitivity function or at least the reference to Griffith et al. (2022). You replied to the corresponding comment by giving detailed information on the sensitive depth ranges related to specific pulse lengths. Please include this information also in the manuscript, at least.

P5L117: As stated later in the manuscript, the water tables in the boreholes have been acquired up to a few decades ago. It is necessary to mention this important detail already here in the Section "Methods" together with a statement on the plausibility of comparing this data with the recent estimates from SNMR.

P5L121: Please give information on min and max values of layer thicknesses in the kernel.

P6L145: "north-east" > Seems to be "north-west" according to Fig. 2a.

P6L152: "The elevation varies…" Reformulation is necessary.

P6L152: "The middle field…" Sentence is redundant and unnecessary.

P6L156: "The data…" Reformulation is necessary. SNMR measures water content instead of pressure head. "Rather" is the wrong word in this context.

P7L157: "The borehole…" > "Borehole data in this area…"

P7L171: Please comment on the fact that this underlying till is obviously not resolved by the TEM data.

P7L185: Include Grombacher et al. (2022) here as a reference for this procedure.

P9L188: Fig.3a shows a decrease of elevation of water tables towards west, not north.

P9L191: "borehole data has been acquired …"

P9L194: Not "uniform" at all, the water table varies within 10 m from east to west.

P9L104: Reformulation is necessary. I do not understand what this statement is pointing to.

P11L218: Please reformulate and clarify, "flow path" is not the correct feature in this context.

P11L219: "northern most" > "most northern"

P11L223: Please explain: Why did you not use a different resolution?

P11L237: Less saturated or unsaturated conditions are not possible beneath the water table. An increasing clay content is the only reliable explanation.

P11L239: wrong word in this context: "perturb" (the same for P16L309)

P12L244: Reformulation is necessary: "Therefore,…"

P12L251-257: In other words, the comparison of TEM and NMR in Fig.5b is meaningless. As already mentioned in the first review of this manuscript, it is not necessary to show this data. We do not learn anything from it.

P13L274: "decrease amount of…" > "decreased number of…"

P15L284: Reformulation and clarification is necessary: clayey layers are conductive but most likely appear with high water content in reality. However, we do not see this clay-bound water with SNMR.

P15L286: As you explained in the reply of my first review, such development is planned for future research. You should give this information also here as an outlook.

P15L292: There are some examples in the literature demonstrating that this combination is indeed promising. Please cite at least one or two of them here.

P16L296: I question this statement. Some water table estimates in your study are consistent with the borehole data, and some are not. I acknowledge the explanations and discussion on the differences, but the terminus "consistence" points to a conclusion that cannot be given at this state for various reasons.

First of all, as you also mention at some point in the manuscript, the water table in boreholes is actually a pressure head that, from a physical viewpoint, cannot be in consistence with the elevation of the saturated zone that is measured by SNMR – even for unconfined aquifers you have to consider the capillary fringe. The difference might irrelevant in your areas but we do not know for sure. Second, we are not yet able to identify the confidence bounds of the water table estimates from SNMR. As you explained in your reply on the first review, the corresponding analysis is still ongoing and I am very curious about it. Last but not least, your data does not show ground truth, because there are years and decades between borehole data acquisition and the NMR measurements.

---

## Author Response (AR2)

Review of manuscript hess-2022-356 (revised version)

**Technical note: High density mapping of regional groundwater tables with steady-state surface nuclear magnetic resonance – three Danish case studies**

**General assessment**

The authors touched many aspects of the first review in their revision. By consequently showing the elevation of water tables instead of depth below surface as in the first version, the estimates from the SNMR data can be assessed as being much more plausible than before. The discussion of the results has been extended by introducing possible reasons for the remaining discrepancies between the different data sets.

However, there are still inconsistencies that need to be clarified, and, unfortunately, the manuscript still has significant linguistic deficits that have to be corrected before publication. Thus, I suggest moderate revisions.

Author response:
We thank the reviewer for the detailed comments. Below is a detailed description of changes made to the manuscript.

**Details**
The page and Line numbers refer to the document with marked changes (hess-2022-356-ATC1_comments.pdf)

P4L96: The sentence "Furthermore,…" can be erased or should be reformulated.

Author response: Deleted redundant sentence.

P4L97: with > at

Author response: Changed "*with*" to "*at*" in P4L96.

P4L102: redundant information

Author response: Deleted sentence

P4L103: What do you mean by "having relaxation time and spatial overlapping"? A tradeoff between resolving spatial information and discovering the relaxation times of the signals? Please clarify.

Author response: Thank you for the comment. We have added a description of how this statement should be understood and clarified the sentence. As mentioned in later comments, by increasing the pulse length the penetration depth increases. If we change repetition times instead, we can better resolve the relaxation time, more explanation in Griffiths et al., (2022). More information is added is added to the manuscript in P4101-103.

The sentence has been clarified from:

*"The pulse protocols and current amplitudes were chosen based on having relaxation time and spatial sensitivity."*
P4L101-104
To:
*"Pulse protocols and current amplitudes are varied to encode both spatial and relaxation time information in the collected data set. Variable current amplitudes are used to manipulate the depths of origin of the signal, while relaxation time information is encoded through manipulation of the repetition time. This is because varying the repetition time alters the induced steady-state amplitude, which is based on the underlying relaxation times (Griffiths et al., 2022)."*

P4L101-103: As already suggested in my first review, I would prefer a figure with an example of a sensitivity function or at least the reference to Griffith et al. (2022). You replied to the corresponding comment by giving detailed information on the sensitive depth ranges related to specific pulse lengths. Please include this information also in the manuscript, at least.

Author response: We have added the information to the manuscript with the sentence above including these details.

P5L117: As stated later in the manuscript, the water tables in the boreholes have been acquired up to a few decades ago. It is necessary to mention this important detail already here in the Section "Methods" together with a statement on the plausibility of comparing this data with the recent estimates from SNMR.

Author response: We have added a few sentences at the end of the method section to clarify the plausibility of the borehole measurements.

P5L116-119:
*"Water table measurements range from a year to several decades old. It is plausible that these water tables have varied considerably by extraction. However, the consistency of these water table measurements across the borehole database suggests a relatively stable system throughout the years. By reproducing water table estimates consistent with available borehole data, we demonstrate the ability of surface NMR to reliably estimate the water table surface."*

P5L121: Please give information on min and max values of layer thicknesses in the kernel.

Author response: Added the minimum and maximum of layer thicknesses and now reads:
P5L122:
*"The kernels are discretized by a 26-layer model with increasing thickness at depth from 0.5m to 5.0m, to a total depth of 50m."*

P6L145: "north-east" > Seems to be "north-west" according to Fig. 2a.

Author response: Changed in P6L143 from "*north-east*" to "*north-west*".

P6L152: "The elevation varies…" Reformulation is necessary.

Author response: The sentence has been reformulated from:

*"The elevation of the water table varies 12m in the area."*

To P6149-150:
*"The water table elevation ranges from 11m to 22m in the area."*

P6L152: "The middle field…" Sentence is redundant and unnecessary.

Author response: The sentences have been deleted.

P6L156: "The data…" Reformulation is necessary. SNMR measures water content instead of pressure head. "Rather" is the wrong word in this context.

Author response: The sentence has been deleted and replaced. From:
*"The data can also be explained if the aquifer is confined because the SNMR data is identifying where the water resides rather than the pressure head."*

To P6154-156:
*"The SNMR results identify the physical location of water at depth and not the hydraulic head, as such if the aquifer is confined, there will be differences between SNMR estimated water tables and the pressure head from wells. "*

P7L157: "The borehole…" > "Borehole data in this area…"

Author response: Changed to read P6L156:
*"Borehole data in this area.."*

P7L171: Please comment on the fact that this underlying till is obviously not resolved by the TEM data.

Author response: A comment on the TEM data here is added. On inspection the TEM identifies resistivities of 90 ohmm to 60 ohmm which is in the range of Danish tills. It does not, however, resolve the upper high resistive sand which is an extremely difficult target for tTEM as it is very shallow and resistive.

Added P7166-167:
*"The TEM profile identifies a layer of 90 ohmm to 60 ohmm which is consistent with Danish tills. The shallow resistive sand layer is difficult to resolve with TEM."*

P7L185: Include Grombacher et al. (2022) here as a reference for this procedure.

Author response: Added the reference in P7181.

P9L188: Fig.3a shows a decrease of elevation of water tables towards west, not north.

Author response: Added west, but there is still a considerable change in water table towards the north and will now read:
P9184-185
*"A slight decrease in elevation of the water table is visible towards the north-west part of the area.."*

P9L191: "borehole data has been acquired …"

Author response: Fixed.

P9L194: Not "uniform" at all, the water table varies within 10 m from east to west.

Author response: Sentence has been deleted.

P9L104: Reformulation is necessary. I do not understand what this statement is pointing to.

Author response: The sentence was reformulated from:

"The duality of $T_2^*$ and water contents could explain some of these consistencies and a limitation of only using water contents as a mean of identifying aquitards."

To P9L197-198:
"As there is little data influence, the data is fitted without altering the water content from the starting model of 10% in S6. In general, …"

P11L218: Please reformulate and clarify, "flow path" is not the correct feature in this context.

Author response: The statement was changed from "flow path" to "flow direction" in P11212

P11L219: "northern most" > "most northern"

Author response: Changed

P11L223: Please explain: Why did you not use a different resolution?

Author response: A clarification is added here to explain why this resolution is not changed. The sentences are changed from:
"The resolution of the model is an important aspect for determining the water table, and if targets are generally deeper, a different resolution could be used. This is not the case here where most of the water table estimations is ranging from 5~m to 10~m. Similarly, the eastern most sounding have a similar effect."

To P11L215-218:
"Similarly, the eastern most sounding have a similar effect. The discretization of the model is an important aspect of estimating the layer thicknesses. The discretization reflects the decrease in sensitivity with depth and adding more layers would make the inversion more regularized. As most of the water table depths are 5~m to 10~m these issues are not as profound."

P11L237: Less saturated or unsaturated conditions are not possible beneath the water table. An increasing clay content is the only reliable explanation.

Author response: Thank you for the comment. We have reformulated and emphasized that it would be an increase in clay content.
P11L224-226

*"The conductive unit coincides with a decrease in water content at 45~m to 55~m elevation for S5, S7, and S8. By the SNMR results alone, the decrease could indicate a unit containing more bound water, i.e., an increase in clay content."*

P11L239: wrong word in this context: "perturb" (the same for P16L309)

Author response: In P11L227 has been changed to "*identify*".
In P16L293 to "*investigated*".

P12L244: Reformulation is necessary: "Therefore,…"

Author response: The sentence has been clarified from:
*"Therefore, most of the discrepancy at this location is likely due to how the SNMR and TEM results being placed several hundreds of meters away from each other."*

P12L233-234
*"The projection of SNMR water contents onto the TEM profile is likely the reason for this inconsistency."*

P12L251-257: In other words, the comparison of TEM and NMR in Fig.5b is meaningless. As already mentioned in the first review of this manuscript, it is not necessary to show this data. We do not learn anything from it.

Author response: As the TEM is the only other data available, we think that it is important to show this data and the influence that the projection might have on the results. We have changed the description of these inconsistencies and point to the fact that these effects often occur when dealing with several data types.

Changed from:
*"Since the profile is located along a TEM line, the SNMR water content profiles are projected about 100~m to 200~m onto the resistivity profile. The geology changes quite rapidly in these glacial landscapes, which could explain parts of these differences. Additionally, TEM and SNMR arises from different geophysical phenomena, which implies that a change in water content is not necessarily seen in the resistivity profile and vice versa."*

P12L238-239
*"SNMR soundings projected 100m to 200m could measure a different subsurface as changes in geological conditions may occur at these length scales in glacial landscapes."*

P13L274: "decrease amount of…" > "decreased number of…"

Author response: Fixed.

P15L284: Reformulation and clarification is necessary: clayey layers are conductive but most likely appear with high water content in reality. However, we do not see this clay-bound water with SNMR.

Author response: We have clarified the sentence adding that it would be low free water units for the SNMR. From:
*"Furthermore, comparison with the high spatial coverage of tTEM showed good agreement in finding conductive and low water bearing units."*

*To P15L266-267:*
*"Furthermore, comparison with the high spatial coverage of tTEM showed good agreement in finding conductive units as low free water units for the SNMR."*

P15L286: As you explained in the reply of my first review, such development is planned for future research. You should give this information also here as an outlook.

Author response: Added an outlook to the end of this sentence.
*"… and will be investigated further in future research."*

P15L292: There are some examples in the literature demonstrating that this combination is indeed promising. Please cite at least one or two of them here.

Author response: Added two references

P13L275-276:
Irons, T.P., Martin, K.E., Finn, C.A., Bloss, B.R. and Horton, R.J., 2014. Using nuclear magnetic resonance and transient electromagnetics to characterise water distribution beneath an ice covered volcanic crater: The case of Sherman Crater Mt. Baker, Washington. *Near Surface Geophysics*, *12*(2), pp.285-296.

Behroozmand, A.A., Auken, E., Fiandaca, G. and Christiansen, A.V., 2012. Improvement in MRS parameter estimation by joint and laterally constrained inversion of MRS and TEM data. *Geophysics*, *77*(4), pp.WB191-WB200.

P16L296: I question this statement. Some water table estimates in your study are consistent with the borehole data, and some are not. I acknowledge the explanations and discussion on the differences, but the terminus "consistence" points to a conclusion that cannot be given at this state for various reasons.

Author response: Thank you for your comment. We generally use "consistent" when the majority of measurements are comparable with other data such as boreholes or TEM data.
We have changed the sentence to read "consistent with most borehole".

P15L279-280
*"However, the water level estimation by the largest gradient in the water content profile has been consistent with most borehole measurements."*

First of all, as you also mention at some point in the manuscript, the water table in boreholes is actually a pressure head that, from a physical viewpoint, cannot be in consistence with the elevation of the saturated zone that is measured by SNMR – even for unconfined aquifers you have to consider the capillary fringe. The difference might irrelevant in your areas but we do not know for sure. Second, we are not yet able to identify the confidence bounds of the water table estimates from SNMR. As you explained in your reply on the first review, the corresponding analysis is still ongoing and I am very curious about it. Last but not least, your data does not show ground truth, because there are years and decades between borehole data acquisition and the NMR measurements.

Author response: A comment on capillary fringe is added. Since the capillary fringe in sand aquifers are limited to below 1m it is hard to resolve these differences in the model.

*"Another aspect is that the water table estimate from the SNMR includes the capillary fringe. However, the difference would be limited in this study as all aquifers are sand aquifers and would have a small capillary fringe (Bevan et al., 2005) compared to the discretization of the model."*

We agree that the confidence bounds are a very important aspect of these estimates. It is difficult to estimate uncertainties with the regularized deterministic inversion. The stochastic inversion results will be interesting to see how these confidence bounds vary.

We have added a description of the uncertainties in using boreholes for comparisons. We agree that the borehole data is not necessarily ground truth, but the borehole data base represents our best approximation of ground truth. Despite the time periods between the different borehole observations of water table, they remain consistent with one another giving confidence that we can still use these data for comparison with the SNMR results.

---

## Author Response (AR3)

Review of manuscript hess-2022-356 (revised version)

**Technical note: High density mapping of regional groundwater tables with steady-state surface nuclear magnetic resonance – three Danish case studies**

**General assessment**

Both reviewers provided positive assessment of the present version of the manuscript, which now requires only minor corrections.

I ask the authors to consider the technical comments by Anonymous referee #1 and to comply with the data policy of EGU journals, as described in https://www.hydrology-and-earth-system-sciences.net/policies/data_policy.html. In particular, "Copernicus Publications requests depositing data that correspond to journal articles in reliable (public) data repositories" and if this not possible, a proper justification should be given.

Author response:
We thank the reviewer for the suggested corrections. The changes to the manuscript are highlighted below. As the data is stored in files only accessible with a certain python environment not readily available, all data is available upon request to the corresponding author.

P5L119 (manuscript with marked changes): "… range from a year to … old" => "… are one year to … old"

Author response: We changed the sentence following the comment.

P11L224 (manuscript with marked changes): "…eastern most…" => "…most eastern…" or "easternmost"

Author response: P11L224 now reads: "*..most eastern..*".

P12L253: I suggest a simpler formulation, maybe: "TEM profile and SNMR sounding points are 100 m to 200 m apart and may thus probe a different subsurface…"

Author response: The sentence has been changed from
P12L253:
"*SNMR soundings projected 100m to 200m could measure a different subsurface as changes in geological conditions may occur at these length scales in glacial landscapes.*"
To:
"*The TEM profile and SNMR soundings are 100~m to 200~m apart and may thus probe a different subsurface in this glacial landscape.*

---

## Author Response (AR4)

**Technical note: High density mapping of regional groundwater tables with steady-state surface nuclear magnetic resonance – three Danish case studies**

**General assessment**

The paper is almost ready to be accepted for publication, but the data availability issue must be fixed.

Namely, the answer by the authors that "the data is stored in files only accessible with a certain python environment not readily available" means that the FAIR concept is not fulfilled.

I recall that "Copernicus Publications requests depositing data that correspond to journal articles in reliable (public) data repositories". Therefore, I ask authors to make their data set FAIR, in full compliance with HESS data policy (https://www.hydrology-and-earth-system-sciences.net/policies/data_policy.html).

As a marginal option, this policy requires that "if the data are not publicly accessible, a detailed explanation of why this is the case is required". In the specific case, it would be necessary to specify which python environment is required to read the data and which data format can be used to share data with interested researchers.

Please, if the same issue (restrictions due to the development environment) applies to the codes, this should be explicitly stated in the "Code and data availability" section.

Author response:
We have deposited the data from the survey in an online repository at Zenodo. Here, all TEM and SNMR inversions used for figures are available.
As for the code, the python package is called Apsu, an in-house developed package, and is not available as it is largely in development with no stable release.

P16 L315

From:

*"Code and data availability. Code and data are available upon request to the corresponding author."*

To:

*"Data availability. All SNMR and tTEM data in this survey is available at https://doi.org/10.5281/zenodo.8186351."*